# ReAugment: Targeted Few-Shot Time Series Augmentation via Model Zoo-Guided Reinforcement Learning

**Haochen Yuan** [1]  **Yutong Wang** [1]  **Yihong Chen** [1]  **Yunbo Wang** [1]  **Xiaokang Yang** [1]

## Abstract

Few-shot time series forecasting suffers from severe overfitting due to limited high-quality training data. We introduce **ReAugment**, a reinforcement learning (RL) framework that explicitly learns *where* and *how* to augment time series data. ReAugment maintains a zoo of forecasting models and measures prediction diversity across them to identify training samples that are most prone to model overfitting. These samples are "*bottlenecks*" for generalization and are used as anchor points in the augmentation process. We then employ an RL approach to learn data transformation policies, using a model zoo-guided reward function to bias the transformed data to **overfit-prone regions** of the training distribution that are most beneficial for generalization. A key advantage of the RL formulation is that it avoids backpropagating gradients through the forecasting models, thereby mitigating gradient vanishing. Empirical results across various benchmarks show that ReAugment consistently improves forecasting accuracy in both few-shot and standard settings.

## 1. Introduction

Time series forecasting plays a pivotal role in numerous real-world applications. In recent years, deep learning architectures (Wu et al., 2021; Nie et al., 2023; Liu et al., 2024b) have significantly advanced the state of the art by capturing complex temporal dependencies. However, the success of these models heavily relies on the availability of large-scale, high-quality labeled data. In many practical scenarios, we are often confronted with the few-shot forecasting problem, where only a limited number of historical observations are available. Under such data-scarce conditions, deep models

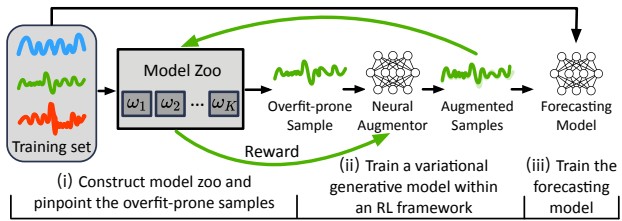

*Figure 1.* **Systematic overview of ReAugment.** We propose a reinforcement learning-based closed-loop framework tailored for few-shot time series forecasting. The pipeline follows an integrated "*identify-augment-forecast*" cycle, where overfit-prone samples are dynamically targeted and augmented to enhance generalization.

are notoriously prone to **severe overfitting**, leading to poor generalization on unseen future sequences.

Data augmentation is a promising strategy to mitigate data scarcity. Traditional techniques (Wen et al., 2021; Cheung & Yeung, 2020), such as adding Gaussian noise, scaling, or magnitude warping, primarily rely on manual heuristic designs. While helpful, these "static" augmentation methods often fail to account for the underlying distribution of the specific dataset. Recently, automated augmentation frameworks have attempted to learn optimal policies (Demirel & Holz, 2024; Schneider et al., 2024). However, these methods are generally **NOT** task-oriented, as the augmentation process is decoupled from the performance of downstream forecasting models.

In this paper, we argue that *closed-loop data augmentation* is essential, in which the objectives of data augmentation should be directly aligned with forecasting performance. We present ReAugment, a reinforcement learning (RL)-based data augmentation framework that explicitly learns *where* and *how* to augment time series data. Our core insight is that samples where different forecasting models disagree (*i.e.*, high prediction diversity) are often those most prone to overfitting. Based on this, ReAugment maintains a **model zoo** (*i.e.*, a collection of diverse forecasting architectures) to evaluate the uncertainty and "overfit-propensity" of training samples. These critical samples identify which specific distribution areas are the "bottlenecks" for generalization and are designated as *anchor points* for augmentation.

We formulate the augmentation process as an RL task, where parameterized transformation policies are learned condi-

[1] MoE Key Lab of Artificial Intelligence, Institute of AI, School of Computer Science, Shanghai Jiao Tong University, Shanghai, China. Correspondence to: Yunbo Wang <yunbow@sjtu.edu.cn>.

*Proceedings of the 43$^{rd}$ International Conference on Machine Learning*, Seoul, South Korea. PMLR 306, 2026. Copyright 2026 by the author(s).

tioned on the anchor points. To guide the policy, we define a model zoo-based reward function that balances diversity and fidelity: It encourages generating samples in regions **prone to model overfitting** while preserving consistency with the original data distribution. While using the forecasting performance as reward signals, the RL formulation allows the framework to optimize the policy without backpropagating gradients through the entire forecasting model zoo, effectively bypassing the gradient vanishing problem.

The contributions of this work are summarized as follows:

- We propose ReAugment, a closed-loop method that adaptively identifies and augments critical samples to improve generalization in few-shot time series forecasting.

- We introduce a model zoo-guided mechanism to locate anchor points within the training distribution that are most susceptible to model overfitting.

- We design a policy optimization method that uses forecasting performance as a reward, ensuring the generated data effectively covers "hard" regions while remaining computationally efficient and gradient-friendly.

- Extensive experiments on multiple real-world datasets demonstrate that ReAugment consistently outperforms existing augmentation methods across various forecasting architectures, showing significant gains in both few-shot and standard forecasting scenarios.

## 2. Related Work

**Transformer-based time series forecasting.** Deep learning has revolutionized time series forecasting, evolving from early CNN and RNN-based architectures (Torres et al., 2021; Wang et al., 2022; Che et al., 2018; Sagheer & Kotb, 2019) to the current dominance of Transformer-based models (Li et al., 2019; Wu et al., 2021; Zhou et al., 2021; Liu et al., 2021; Zhou et al., 2022; Zhang & Yan, 2022; Zeng et al., 2023; Cao et al., 2024; Yi et al., 2024). Recent advancements have focused on capturing long-range dependencies through innovative attention mechanisms. For example, PatchTST (Nie et al., 2023) adopts a patch-wise approach to enhance local semantic extraction and reduce computational complexity, while iTransformer (Liu et al., 2024b) inverts the traditional structure to learn variate-wise representations, achieving state-of-the-art results on multivariate datasets. Despite these architectural breakthroughs, even the most powerful Transformers remain vulnerable to overfitting when training data is limited.

**Few-shot time series forecasting.** In many practical scenarios, acquiring large-scale, high-quality historical data is hindered by collection costs or the inherent scarcity of new events. To address this, few-shot learning paradigms have been explored, ranging from prior-data fitted networks to specialized spatiotemporal architectures (Dooley et al.,

2023; Xu et al., 2024; Jiang et al., 2023; Yuan et al., 2024). More recently, the field has shifted the focus toward large-scale time series foundation models (Das et al., 2024; Jin et al., 2024; Bian et al., 2024; Ekambaram et al., 2024; Liu et al., 2024a;c; Pan et al., 2024), which leverage massive pre-training for zero-shot generalization. However, these models often exhibit a critical fragility: as the distribution gap between the pre-training and target data widens, their generalization performance degrades significantly.

**Time series augmentation.** Data augmentation techniques for time series can be broadly categorized into three categories (Iglesias et al., 2023). The first involves heuristic transformations in the time, frequency, or magnitude domains, such as slicing (Cao et al., 2020), frequency warping (Cui et al., 2015), and jittering (Flores et al., 2021). The second category contains the learning-based methods, including contrastive learning (Demirel & Holz, 2024), and Mixup variants (Schneider et al., 2024). The third tier utilizes generative models, such as GANs (Yoon et al., 2019; Liao et al., 2020), VAEs (Sohn et al., 2015; Li et al., 2020), and Diffusion models (Huang et al., 2023), to synthesize realistic sequences.

While RL-based sample selection has been explored in other domains to choose between predefined operations (Huang et al., 2024; Yang et al., 2025), it remains under-explored in the context of time series. We present a pioneering approach that applies RL to optimize a generative augmentation policy within a continuous latent space. This allows our agent to go beyond selecting discrete operations, instead performing fine-grained, task-aware perturbations that directly target the forecasting model's generalization bottlenecks.

## 3. Anchor Points Identification

### 3.1. Identifying Overfit-Prone Data with a Model Zoo

In few-shot time series forecasting, a critical observation is that forecasting models do not uniformly overfit the training set. Instead, they tend to overfit specific samples or regions. Intuitively, these overfit-prone data represent the bottleneck in the model's generalization capability. By generating synthetic samples in nearby data distributions, we can effectively force the model to learn more robust temporal patterns. Therefore, the primary challenge lies in quantifying this overfit-propensity to select appropriate anchor points for the subsequent generative augmentation process.

**Formulating the model zoo.** To robustly identify these samples, we use a cross-validation-based ensemble, termed a *model zoo*. We partition the training set $\mathcal{D}$ into $K$ disjoint folds $\{\mathcal{D}_1, \mathcal{D}_2, \ldots, \mathcal{D}_K\}$. For each fold $\mathcal{D}_k$, we train a forecasting model $f_{\omega_k}$ on the remaining $K-1$ folds $\mathcal{D} \setminus \mathcal{D}_k$. This yields a model zoo $\mathcal{M} = \{f_{\omega_1}, f_{\omega_2}, \ldots, f_{\omega_K}\}$, where each model has a unique "blind spot" regarding one specific subset of the data.

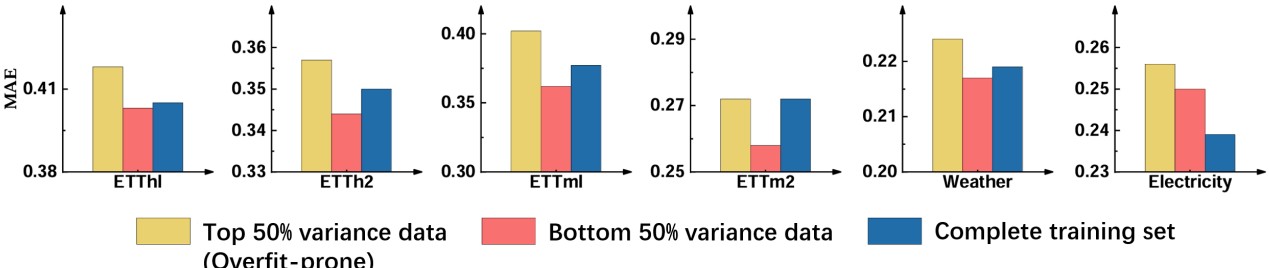

*Figure 2.* **Preliminary findings on overfit-prone data.** We compare the performance of *iTransformer* (Liu et al., 2024b) trained with different splits of the original training set, which are divided based on the variance of prediction errors across the forecasting model zoo.

**Quantifying overfit-propensity via model-zoo variance.** For any training sample $(x, y) \in \mathcal{D}_k$, we define its overfit-propensity by measuring the discrepancy between models that have seen the data and those that have not. Specifically, let $\mathcal{L}(f(x), y)$ denote the prediction error (*e.g.*, MSE). For a sample $x \in \mathcal{D}_k$, we compute the error set $E_x = \{e_1, e_2, \ldots, e_K\}$, where $e_i = \mathcal{L}(f_{\omega_i}(x), y)$. We then define the Model-Zoo Variance $\mathcal{V}(x)$ as:

$$\mathcal{V}(x; \mathcal{M}) = \frac{1}{K} \sum_{i=1}^{K} \left( e_i - \bar{E}_x \right)^2. \quad (1)$$

where $\bar{E}_x = \frac{1}{K} \sum_{i=1}^{K} e_i$. A high $\mathcal{V}(x)$ indicates that the prediction for $x$ is highly sensitive to the specific training distribution the model was exposed to. Such samples are typically located in regions where the model's *empirical risk minimization* (ERM) fluctuates significantly, signaling a high risk of overfitting. We rank all training samples by $\mathcal{V}(x)$ and select the top $p\%$ (*e.g.*, $50\%$) as anchor points $\mathcal{A} \subset \mathcal{D}$ for augmentation.

### 3.2. Empirical Evidence

To empirically validate whether $\mathcal{V}(x)$ effectively identifies the "bottleneck" of generalization, we conduct a controlled experiment on standard benchmarks including *ETT*, *Weather*, and *Electricity*. We divide the training set into two equal-sized groups based on the Model-Zoo Variance: Group A (Top $50\%$, High Variance) and Group B (Bottom $50\%$, Low Variance). We then train two identical iTransformer models independently on each group and evaluate them on the same unseen test set.

> **Key Insight**
>
> The samples in Group A act as a "generalization bottleneck." Their high variance indicates that models struggle to converge to a consistent solution in these regions, leading to poor out-of-distribution performance.

As illustrated in Figure 2, the performance gap is striking: models trained exclusively on Group B (the "stable" sam-

ples) consistently achieve significantly lower MAE/MSE across all datasets compared to those trained on Group A.

This finding provides a principled justification for our approach: rather than augmenting the entire dataset, which might introduce redundant information from the already stable Group B, ReAugment focuses its capacity on the high-variance regions (Group A). By generating "difficult but realistic" variants around these anchor points, we specifically target the distribution gaps that hinder the model's ability to generalize.

## 4. Closed-Loop Augmentation

The core of ReAugment is an "identify-augment-forecast" closed-loop system. Building upon our preliminary findings, we propose to treat data augmentation as a targeted learning process: We first identify the "generalization bottlenecks" (overfit-prone samples) and then deploy a generative agent to explore and cover the high-variance regions surrounding these anchor points.

### 4.1. Overall Training Pipeline

The training workflow of ReAugment is structured into three distinct stages (detailed in Algorithm 1 in the appendix):

- **Stage A: Anchor points identification.** We first pinpoint the "generalization bottlenecks" of the training set with the Model-Zoo Variance, forming a targeted subset $\mathcal{D}_s$ that is most susceptible to model overfitting.

- **Stage B: Generative initialization.** We train a Variational Masked Autoencoder (VMAE) on the identified overfit-prone samples. By reconstructing masked time series sequences conditioned on absolute timestamps, the VMAE learns a robust base distribution, providing a stable starting point for the RL agent.

- **Stage C: Policy finetuning via RL.** We treat the VMAE prior as a policy and finetune it using Group Relative Policy Optimization (GRPO). The agent learns to navigate the latent space to generate samples that maximize a reward function reflecting both prediction diversity (from the model zoo) and data fidelity.

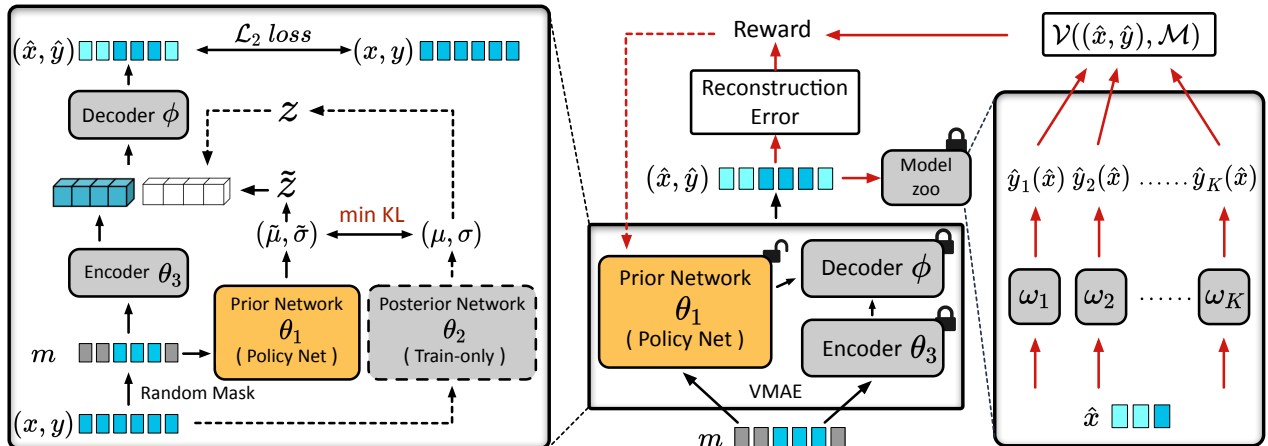

*Figure 3.* **Model details. Left:** ReAugment pretrains a Variational Masked Autoencoder (VMAE) to capture the intrinsic distribution of overfit-prone anchor points. **Right:** An RL framework finetunes the VMAE prior network using GRPO. The latent space serves as the action space, while reward signals from the forecasting model zoo guide the generation of diverse, generalization-enhancing samples.

This staged approach ensures that the augmentor does not merely replicate the training data but actively fills the gaps where the forecasting model is most vulnerable.

### 4.2. Variational Masked Autoencoder as the RL Actor

To generate high-quality, diverse time series data, we adopt a probabilistic generative framework. The neural augmentor can be denoted as $(\hat{x}, \hat{y}) \sim G((x, y), z)$, where $z$ is a latent variable. Our objective is to learn a latent distribution that balances diversity (to cover unseen patterns) and fidelity (to remain within the physical constraints of the data).

As illustrated in Figure 3 (left), the VMAE comprises four functional modules:

1. **Prior Network ($p$):** Estimates the distribution of $z$ based on masked data $m$ and absolute timestamps $t$.
2. **Posterior Module ($q$):** Refines the distribution of $z$ using the full sequence $(x, y)$.
3. **Data Encoder (Enc):** Extracts deep temporal features from the masked input.
4. **Decoder (Dec):** Projects the combined features and latent samples back to the original data space.

The components are formulated as:

$$
\begin{aligned}
\text{Prior}: \quad & \tilde{z} \sim p(m, t; \theta_1), \\
\text{Posterior}: \quad & z \sim q((x, y), t; \theta_2), \\
\text{Encoder}: \quad & u = \text{Enc}(m, t; \theta_3), \\
\text{Decoder}: \quad & (\hat{x}, \hat{y}) = \text{Dec}(\text{concat}(u, z); \phi).
\end{aligned}
\tag{2}
$$

During Stage B, we minimize the Evidence Lower Bound (ELBO), combining a reconstruction $\mathcal{L}_2$ loss with a Kullback–Leibler (KL) divergence constraint:

$$
\begin{aligned}
\mathcal{L} = \; & \mathbb{E}_{(x,y) \sim D_s} \|(\hat{x}, \hat{y}) - (x, y)\|_2^2 \\
& + \beta \, \mathcal{D}_{KL}(q(z \mid (x, y), t) \;\|\; p(\tilde{z} \mid m, t)),
\end{aligned}
\tag{3}
$$

where $\mathcal{D}_s$ denotes the set of overfit-prone anchor points. In general, the VMAE design can be integrated into any encoder-decoder-based time series forecasting architecture. In this work, we specifically adopt the encoder and decoder (implemented as a linear projector) from iTransformer (Liu et al., 2024b) for feature extraction and decoding.

### 4.3. GRPO Guided by Model-Zoo Variance

Standard backpropagation through the forecasting model zoo is often plagued by gradient vanishing or instability due to the complex, non-linear landscape of pretrained model parameters. To address this, we reformulate the finetuning of the neural augmentor as a one-step contextual bandit. For each episode:

- **State:** A masked time series $m$ with its timestamp $t$.
- **Action:** A latent $z$ sampled from the prior $\pi_{\theta_1}(z \mid m, t)$.
- **Reward:** A scalar feedback derived from the model zoo's predictive discrepancy.

This formulation avoids the complexity of temporal credit assignment ($\gamma = 0$) while focusing the agent on the quality of immediate data generation.

**Reward design.** The reward function is designed to incentivize the generation of samples that are "difficult" (high model-zoo variance) yet "realistic" (low reconstruction error). Specifically:

$$
\begin{aligned}
r &= \frac{1}{1 + e^{-\eta \cdot f(\hat{x}, \hat{y})}}, \\
\text{where} \quad f(\hat{x}, \hat{y}) &= \frac{\mathcal{V}((\hat{x}, \hat{y}); \mathcal{M})}{\|(\hat{x}, \hat{y}) - (x, y)\|_2^2 + \epsilon}.
\end{aligned}
\tag{4}
$$

The scaled sigmoid function controlled by $\eta$ prevents saturation near 0 or 1, keeping the reward within a stable range, while $\epsilon$ prevents numerical instability.

**GRPO optimization.** To ensure stable policy updates, we optimize the VMAE prior network using Group Relative Policy Optimization (GRPO). For each masked input $(m, t)$, the policy $\pi_{\theta_1}(z \mid m, t)$ samples a *group* of $G$ latent codes $\{z^{(i)} \sim \pi_{\theta_1}(z \mid m, t)\}_{i=1:G}$, each decoded into an augmented sequence whose reward $r^{(i)}$ is computed by Eq. (4). Unlike traditional PPO, GRPO computes a relative advantage without a separate value function:

$$A^{(i)} = \frac{r^{(i)} - \text{mean}(r^{(1:G)})}{\text{std}(r^{(1:G)})}. \tag{5}$$

The policy objective is then defined as:

$$J(\theta_1) = \mathbb{E}_{(m,t)} \left[ \frac{1}{G} \sum_{i=1}^{G} A^{(i)} \log \pi_{\theta_1}(z^{(i)} \mid m, t) \right]. \tag{6}$$

**Why GRPO?** By normalizing rewards within each group, GRPO produces a scale-invariant advantage signal. This is particularly beneficial when dealing with a model zoo, as different forecasting models may produce reward magnitudes of varying scales. GRPO effectively filters out this noise, providing a stable gradient signal that guides the VMAE prior toward the most effective augmentation regions.

## 5. Experiments

### 5.1. Experimental Setups

**Datasets and tasks.** We evaluate ReAugment across five widely recognized real-world time series benchmarks: ETT (four subsets), Traffic, Electricity, Weather, and Exchange. Following standard protocols (Liu et al., 2024b; Nie et al., 2023), we employ a lookback length of 96 and a prediction horizon of 96. We consider two primary settings:

- *Few-shot setup*: To simulate extreme data scarcity, we use only the first 10% or 20% of the training set. This creates a significant distribution shift between the training and test sets.

- *Standard setup*: The model has access to the full training set to evaluate the generalizability of ReAugment beyond few-shot scenarios.

**Baselines and backbones.** We primarily use iTransformer (Liu et al., 2024b) as the forecasting backbone, while also validating transferability on PatchTST (Nie et al., 2023) and DLinear (Zeng et al., 2023). Unless otherwise specified, we use a model zoo consisting of four cross-validation models. We compare ReAugment with the following augmentation methods:

- *Gaussian*: We use traditional data augmentation methods that apply simple transformations, such as adding Gaussian noise to the raw data. This approach enhances the diversity of the training data by adjusting the controllable variances and means of the added Gaussian noise.

- *Convolve*: We employ another traditional data augmentation method based on the Convolve function in the *Tsaug* library (Wen & Keyes, 2019).

- *TimeGAN* (Yoon et al., 2019): TimeGAN data augmentor combines supervised and adversarial objective optimization. Specifically, through a learned embedding space, the network is guided to adhere to the dynamics of the training data during sampling.

- *ADA* (Schneider et al., 2024): The Anchor Data Augmentation (ADA) method improves domain-agnostic Mixup techniques by generating multiple replicas of modified samples through Anchor Regression, which are then used to create additional training data.

### 5.2. Few-Shot Time Series Forecasting Results

**Main results.** Table 1 summarizes the performance under the few-shot regime. ReAugment consistently achieves the lowest MAE and MSE across all eight benchmarks. Notably, on the ETTh2 and Traffic datasets, ReAugment provides substantial error reductions compared to the second-best method, underscoring the effectiveness of targeting overfit-prone regions via RL. The gains are also consistent across datasets with different temporal characteristics, from relatively regular ETT series to higher-dimensional Traffic and Electricity data. In contrast, heuristic perturbations and generic generative baselines do not uniformly improve the

*Table 1.* **Main results for few-shot forecasting.** We employ iTransformer as the backbone model under identical training seeds. For fair comparison, results for all stochastic baselines are averaged over three random runs; corresponding standard deviations are available in Appendix E. ReAugment consistently outperforms existing heuristic and learning-based augmentation techniques.

| Dataset | Original | | Gaussian | | Convolve | | TimeGAN | | ADA | | ReAugment | |
|---|---|---|---|---|---|---|---|---|---|---|---|---|
| | MAE | MSE | MAE | MSE | MAE | MSE | MAE | MSE | MAE | MSE | MAE | MSE |
| ETTh1 | 0.434 | 0.411 | 0.437 | 0.416 | 0.441 | 0.417 | 0.444 | 0.419 | 0.435 | 0.413 | **0.423**±0.002 | **0.402**±0.001 |
| ETTh2 | 0.362 | 0.320 | 0.365 | 0.321 | 0.364 | 0.323 | 0.366 | 0.327 | 0.368 | 0.331 | **0.337**±0.001 | **0.301**±0.001 |
| ETTm1 | 0.440 | 0.470 | 0.438 | 0.469 | 0.426 | 0.479 | 0.430 | 0.483 | 0.429 | 0.484 | **0.411**±0.001 | **0.436**±0.002 |
| ETTm2 | 0.282 | 0.204 | 0.283 | 0.204 | 0.286 | 0.207 | 0.285 | 0.206 | 0.284 | 0.207 | **0.273**±0.001 | **0.194**±0.001 |
| Weather | 0.231 | 0.187 | 0.240 | 0.196 | 0.253 | 0.204 | 0.239 | 0.191 | 0.246 | 0.198 | **0.228**±0.001 | **0.184**±0.001 |
| Electricity | 0.258 | 0.168 | 0.263 | 0.170 | 0.262 | 0.170 | 0.267 | 0.177 | 0.265 | 0.171 | **0.255**±0.001 | **0.165**±0.000 |
| Traffic | 0.318 | 0.466 | 0.319 | 0.467 | 0.320 | 0.463 | 0.315 | 0.449 | 0.318 | 0.456 | **0.291**±0.001 | **0.427**±0.001 |
| Exchange | 0.228 | 0.103 | 0.229 | 0.104 | 0.226 | 0.099 | 0.226 | 0.100 | 0.235 | 0.116 | **0.224**±0.000 | **0.098**±0.000 |

*Table 2.* **Model-agnostic evaluation of ReAugment across diverse architectures.** We assess the impact of our proposed augmentation method on various forecasting backbones under the few-shot setup. Each configuration is evaluated over three independent random seeds. To maintain conciseness, mean values are reported here, while the corresponding standard deviations are detailed in Appendix Table 14.

| Model | ETTh1 MAE | ETTh1 MSE | ETTh2 MAE | ETTh2 MSE | ETTm1 MAE | ETTm1 MSE | ETTm2 MAE | ETTm2 MSE | Weather MAE | Weather MSE | Electricity MAE | Electricity MSE | Traffic MAE | Traffic MSE | Exchange MAE | Exchange MSE |
|---|---|---|---|---|---|---|---|---|---|---|---|---|---|---|---|---|
| PatchTST | 0.458 | 0.446 | 0.367 | 0.321 | 0.428 | 0.457 | 0.276 | 0.199 | 0.232 | 0.189 | 0.295 | 0.200 | 0.327 | 0.541 | 0.226 | 0.103 |
| + ReAugment | 0.438 | 0.430 | 0.347 | 0.305 | 0.404 | 0.432 | 0.267 | 0.194 | 0.225 | 0.187 | 0.276 | 0.186 | 0.315 | 0.520 | 0.221 | 0.099 |
| DLinear | 0.435 | 0.408 | 0.402 | 0.356 | 0.442 | 0.471 | 0.303 | 0.219 | 0.277 | 0.212 | 0.307 | 0.215 | 0.452 | 0.724 | 0.226 | 0.104 |
| + ReAugment | 0.420 | 0.389 | 0.368 | 0.332 | 0.430 | 0.460 | 0.296 | 0.217 | 0.274 | 0.211 | 0.306 | 0.214 | 0.404 | 0.664 | 0.225 | 0.100 |

*Table 3.* **Augmentation benefit analysis via the Recovery Factor ($\mathcal{F}$).** We evaluate relative performance improvements in few-shot settings using the full training set as a reference. Our approach demonstrates superior robustness across diverse datasets. Detailed statistical results, including standard deviations, are available in Appendix Table 16.

| Dataset | Gaussian $\mathcal{F}_{\mathbf{MAE}}$ | Gaussian $\mathcal{F}_{\mathbf{MSE}}$ | Convolve $\mathcal{F}_{\mathbf{MAE}}$ | Convolve $\mathcal{F}_{\mathbf{MSE}}$ | TimeGAN $\mathcal{F}_{\mathbf{MAE}}$ | TimeGAN $\mathcal{F}_{\mathbf{MSE}}$ | ADA $\mathcal{F}_{\mathbf{MAE}}$ | ADA $\mathcal{F}_{\mathbf{MSE}}$ | ReAugment $\mathcal{F}_{\mathbf{MAE}}$ | ReAugment $\mathcal{F}_{\mathbf{MSE}}$ |
|---|---|---|---|---|---|---|---|---|---|---|
| ETTh1 | -10.3% | -20.8% | -24.1% | -25% | -34.5% | -33.3% | -3.4% | -8.3% | **37.9%** | **37.5%** |
| ETTh2 | -25% | -5.3% | -16.7% | -15.8% | -33.3% | -36.8% | -50.0% | -57.9% | **208.3%** | **100.0%** |
| ETTm1 | 3.2% | 0.8% | 22.2% | -7.0% | 15.9% | -10.1% | 17.5% | -10.8% | **46.0%** | **26.4%** |
| ETTm2 | -10.0% | 0.0% | -40.0% | -16.7% | -30.0% | -11.1% | -20.0% | -16.7% | **90.0%** | **55.6%** |
| Weather | -75.0% | -100.0% | -183.3% | -188.9% | -66.7% | -44.4% | -125% | -122.2% | **25.0%** | **33.3%** |
| Electricity | -26.3% | -10.0% | -21.1% | -10.0% | -47.4% | -45.0% | -36.8% | -15.0% | **15.8%** | **15.0%** |
| Traffic | -2.0% | -1.4% | -4.1% | 4.1% | 6.1% | 23.0% | 0.0% | 13.5% | **55.1%** | **52.7%** |
| Exchange | -4.5% | -5.9% | 9.1% | 23.5% | 9.1% | 17.6% | -31.8% | -76.5% | **18.2%** | **29.4%** |

few-shot model, suggesting that simply increasing sample diversity is insufficient when the generated samples are not aligned with the model's generalization bottlenecks.

**Model-agnostic nature.** As shown in Table 2, ReAugment is not tied to a specific architecture. When applied to PatchTST and DLinear, our method consistently lowers the error metrics, demonstrating its effectiveness as a flexible, model-agnostic augmentation tool. The improvements on both a transformer-style model and a linear model further suggest that the identified overfit-prone regions correspond to data-level deficiencies shared across model families.

**Quantifying augmentation benefits.** To better assess the impact of augmentation relative to data scarcity, we define a new metric, the Augmentation Recovery Factor ($\mathcal{F}$):

$$\mathcal{F}_{\text{MSE}} = \frac{1 - \text{MSE}_{\text{augment}} / \text{MSE}_{\text{few-shot}}}{1 - \text{MSE}_{\text{full}} / \text{MSE}_{\text{few-shot}}}. \quad (7)$$

This metric represents the fraction of the performance gap (between few-shot and full-data training) that is "recovered" by the augmentation. A metric value of $100\%$ indicates that the augmented few-shot model has effectively matched the

error of the full-data model. Negative values indicate that the augmentation is harmful. As shown in Table 3, ReAugment achieves significant positive recovery factors across all datasets, whereas several baselines (like ADA or Gaussian) occasionally result in negative values, indicating that they introduce harmful noise that further hinders generalization. Values approaching $100\%$ indicate that augmentation has nearly recovered the full-data reference performance, while values above $100\%$ occur when the augmented few-shot model surpasses the corresponding full-data reference under the same backbone.

### 5.3. Comparison with Foundation Models

While foundation models, such as TimesFM (Das et al., 2024) and Chronos-2, attempt to solve few-shot tasks through massive pre-training, ReAugment takes a data-centric approach. In Table 4, we compare ReAugment (with iTransformer) against these foundation models under the same few-shot setting. Despite having significantly fewer parameters, the ReAugment-enhanced model outperforms Chronos-2 on all five reported datasets and outperforms

*Table 4.* **Performance comparison between ReAugment (with iTransformer) and foundation models.** Results for Electricity, Weather, and Traffic are omitted for TimesFM to ensure a fair evaluation, as these datasets were included in its pretraining corpus.

| Model | ETTh1 MAE | ETTh1 MSE | ETTh2 MAE | ETTh2 MSE | ETTm1 MAE | ETTm1 MSE | ETTm2 MAE | ETTm2 MSE | Exchange MAE | Exchange MSE |
|---|---|---|---|---|---|---|---|---|---|---|
| Chronos-2 | 0.429 | 0.409 | 0.348 | 0.311 | 0.421 | 0.454 | 0.280 | 0.201 | 0.227 | 0.101 |
| TimesFM | 0.433 | 0.410 | 0.351 | 0.317 | **0.396** | **0.389** | 0.287 | 0.206 | 0.247 | 0.114 |
| ReAugment | **0.423** | **0.402** | **0.337** | **0.301** | 0.411 | 0.436 | **0.273** | **0.194** | **0.224** | **0.098** |

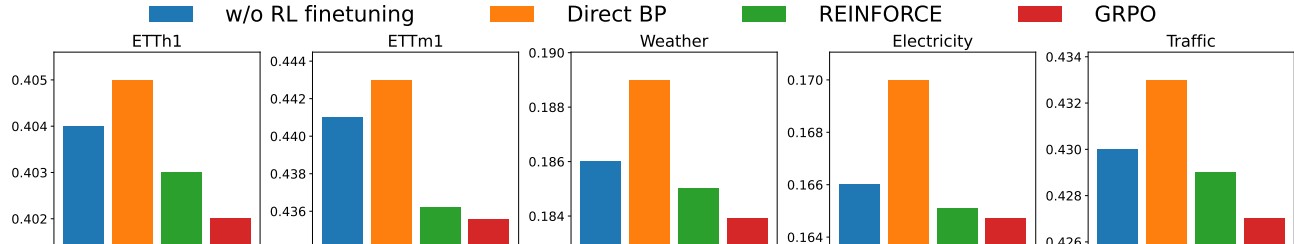

*Figure 4.* **Impact of RL finetuning for few-shot forecasting measured by MSE.** For both variants, the forecasting iTransformer is trained using augmented data three times the size of the original training set.

*Table 5.* **Seen-unseen discrepancy in the model zoo.** We report the average sample-wise MSE of training samples evaluated by models that have seen the corresponding temporal region and by the held-out model that has not. The consistent positive gap supports model-zoo variance as an exposure-sensitivity signal.

| Dataset | Seen MSE | Unseen MSE | Gap |
|---|---|---|---|
| ETTh1 | 0.259 | 0.384 | +48.3% |
| ETTh2 | 0.228 | 0.300 | +31.6% |
| ETTm1 | 0.326 | 0.342 | +4.9% |
| ETTm2 | 0.164 | 0.181 | +10.4% |
| Weather | 0.153 | 0.178 | +16.3% |
| Electricity | 0.120 | 0.149 | +24.2% |
| Traffic | 0.356 | 0.391 | +9.8% |
| Exchange | 0.075 | 0.087 | +16.0% |

*Table 6.* **Effect of error-magnitude filtering within high-variance anchors.** MAE/MSE are reported under the few-shot setting. The full high-variance pool remains consistently competitive, indicating that variance-based exposure sensitivity is not reducible to absolute error magnitude.

| Dataset | High-var + high-error | High-var + low-error | High-var all |
|---|---|---|---|
| ETTh1 | 0.427 / 0.406 | 0.425 / 0.404 | **0.423 / 0.402** |
| ETTh2 | 0.341 / 0.304 | 0.339 / 0.302 | **0.337 / 0.301** |
| ETTm1 | 0.414 / 0.440 | 0.412 / 0.438 | **0.411 / 0.436** |
| ETTm2 | 0.277 / 0.198 | 0.274 / 0.196 | **0.273 / 0.194** |
| Weather | **0.226 / 0.182** | 0.229 / 0.186 | 0.228 / 0.184 |
| Traffic | **0.291** / 0.430 | 0.293 / 0.429 | **0.291 / 0.427** |

TimesFM on four out of five datasets. This suggests that task-specific, targeted augmentation can often be more effective than finetuning large foundation models when domain gaps are large. The comparison also highlights a complementary perspective to scaling: rather than relying solely on broad pretraining coverage, ReAugment explicitly adapts the scarce target-domain training set by synthesizing samples around regions where the forecasting model is most sensitive to data exposure.

### 5.4. Model Analyses

**Ablation of anchor ratios.** Inspired by preliminary findings, we augment the top $50\%$ overfit-prone samples. What would be the impact of augmenting more or fewer samples? To investigate this, we compare augmenting different proportions of overfit-prone samples with augmenting the entire few-shot dataset. For consistency, the total amount of augmented data was kept three times the size of the original training set and applied to the iTransformer model. Figure 5 (left) shows that augmenting the top $50\%$ highest-variance samples yields better results than augmenting the entire dataset ($100\%$ ratio). This confirms our hypothesis that "not all data is created equal" and that focusing generative capacity on generalization bottlenecks is more efficient.

**Seen-unseen discrepancy in the model zoo.** We further examine why model-zoo variance is a meaningful signal for identifying overfit-prone anchors. In our temporally contiguous $K$-fold construction, for each sample in fold $\mathcal{D}_k$, one zoo model has not been exposed to this local temporal region, whereas the remaining models have observed it during training. Table 5 reports the average training-sample MSE evaluated by the seen and unseen models. Across all benchmarks, unseen models consistently incur larger errors, indicating that model-zoo disagreement largely reflects exposure-sensitive generalization gaps rather than arbitrary prediction noise. This supports the use of variance as a diagnostic for locating samples whose local patterns are under-supported in the few-shot training set.

**Variance-based anchor selection.** Beyond varying the number of anchor samples, we further examine whether model-zoo variance provides information beyond simply selecting high-error samples. Within the top $50\%$ high-variance pool, we split anchors by absolute error magnitude and keep the total amount of augmented data unchanged. Table 6 shows that further filtering high-variance anchors by absolute error does not yield consistent gains over using the full high-variance pool. This suggests that the useful signal primarily comes from exposure-sensitive disagreement captured by model-zoo variance, rather than from generic hard-sample selection alone.

**Impact of model zoo size.** We find that increasing the number of models in the zoo generally improves the robustness of the variance signal. As shown in Figure 5 (right),

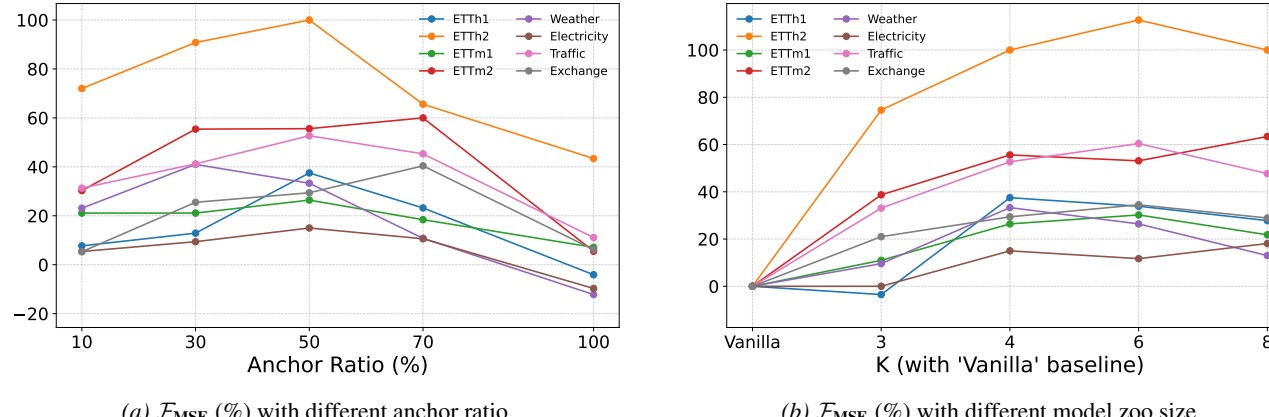

*(a)* $\mathcal{F}_{\mathbf{MSE}}$ (%) with different anchor ratio      *(b)* $\mathcal{F}_{\mathbf{MSE}}$ (%) with different model zoo size

*Figure 5.* **Ablation studies.** (a) Sensitivity analysis regarding the proportion of overfit-prone samples utilized as augmentation anchors. (b) Impact of the model zoo size ($K$) on the Augmentation Recovery Factor ($\mathcal{F}_{\mathbf{MSE}}$). "Vanilla" represents the baseline iTransformer trained without any data augmentation.

performance tends to saturate at $K = 4$, which we adopt as our default setting to balance accuracy and efficiency.

**Rationalizing the prior network as the policy.** We designate the prior network as the policy network primarily because it provides a low-dimensional, regularized latent manifold. This structured action space significantly facilitates stable and efficient RL optimization. In contrast, using the decoder or the full VMAE as the policy would require the agent to navigate a high-dimensional and unconstrained signal space, which typically leads to optimization instability and poor convergence. As demonstrated in Table 7, our prior-based policy consistently outperforms these alternatives across all benchmarks. Optimizing the prior also preserves the decoder as a learned temporal generator, so the RL stage adjusts where to sample in the latent space rather than directly perturbing raw sequences or all reconstruction parameters.

**The role of GRPO.** The RL Stage (Stage B) is critical for moving beyond simple data replication. In Figure 4, we compare our GRPO formulation with: (1) removing RL finetuning entirely, (2) a standard REINFORCE baseline, and (3) direct backpropagation of the same reward function. GRPO provides the most stable performance gains, likely due to its relative advantage signal, which normalizes the diverse reward scales produced by the model zoo. This is useful because reward magnitudes can vary substantially

across datasets and anchor regions; relative normalization makes the policy update less sensitive to such scale differences while retaining the ranking information needed to prefer useful augmentations.

**Computational efficiency.** Despite the use of a model zoo and RL, the training overhead of ReAugment remains manageable. As Table 8 indicates, the total training time for Stages B and C is significantly less than the training time of the forecasting backbone itself. Compared to TimeGAN, which requires a full adversarial training cycle, ReAugment is both more effective and more efficient. Moreover, this cost is incurred only during offline augmentation and training. Once the final forecasting backbone is trained, inference uses the same model architecture as the original baseline and introduces no additional test-time computation.

**Mask rate of VMAE.** We employ a default mask rate of $0.3$ across all experiments in our main analysis. In Table 9, we analyze the sensitivity of this parameter within the few-shot learning framework. The empirical results show that a $0.3$ mask rate provides the best trade-off between generative diversity and reconstruction accuracy, ensuring that the anchor points are augmented with meaningful temporal perturbations. Very small mask rates tend to limit the diversity of reconstructed samples, whereas overly aggressive masking can weaken fidelity to the original temporal structure. The default value therefore encourages useful local variation

*Table 7.* **Comparison of policy network choices in the RL stage.** *Decoder*: Finetuning the VMAE decoder directly; *Full VMAE*: Optimizing all parameters within the VMAE; *Prior* (Ours): Optimizing the prior network to navigate the latent space. Our proposed prior-based policy consistently outperforms other variants by leveraging a more compact and structured action space.

| Policy Net | ETTh1 | | ETTh2 | | ETTm1 | | ETTm2 | | Weather | | Electricity | | Traffic | |
|---|---|---|---|---|---|---|---|---|---|---|---|---|---|---|
| | MAE | MSE | MAE | MSE | MAE | MSE | MAE | MSE | MAE | MSE | MAE | MSE | MAE | MSE |
| Decoder | 0.428 | 0.409 | 0.352 | 0.316 | 0.420 | 0.446 | 0.278 | 0.197 | 0.230 | 0.188 | 0.256 | 0.167 | 0.297 | 0.431 |
| Full VMAE | 0.425 | 0.403 | 0.340 | 0.306 | 0.417 | 0.440 | 0.275 | 0.198 | 0.231 | 0.185 | **0.255** | 0.167 | 0.294 | 0.432 |
| Prior (ours) | **0.423** | **0.402** | **0.337** | **0.301** | **0.411** | **0.436** | **0.273** | **0.194** | **0.228** | **0.184** | **0.255** | **0.165** | **0.291** | **0.427** |

*Table 8.* **Training costs.** The total training time required for augmentation in our method is notably shorter than the time needed for training the forecasting models. We additionally report the training-time overhead of two representative learning-based augmentation baselines, *TimeGAN* and *ADA*. Note that we compare only the training time with other augmentation methods, since the forecasting model training time is comparable across all methods.

| Dataset | TimeGAN | ADA | Our Method | | Forecasting Model | |
| | | | Stage B: VMAE | Stage C: GRPO | iTransformer | PatchTST |
| --- | --- | --- | --- | --- | --- | --- |
| ETTh1 | 4min | 1min | 1min | 2min | 2min | 5min |
| Electricity | 1h 12min | 18min | 24min | 39min | 1h 33min | 2h 24min |
| Traffic | 3h 48min | 58min | 1h 22min | 2h 14min | 4h 27min | 6h 50min |

*Table 9.* **Impact of MAE masking ratio.** The mask rate of 0.3 (our default) generally achieves superior results by providing effective structural perturbations for overfit-prone anchor points.

| Mask rate | 0 | | 0.1 | | 0.3 | | 0.5 | | 0.7 | |
| | MAE | MSE | MAE | MSE | MAE | MSE | MAE | MSE | MAE | MSE |
| --- | --- | --- | --- | --- | --- | --- | --- | --- | --- | --- |
| ETTh1 | 0.437 | 0.410 | 0.429 | 0.408 | **0.423** | **0.402** | 0.426 | 0.405 | 0.433 | 0.414 |
| ETTh2 | 0.361 | 0.321 | 0.354 | 0.313 | **0.337** | **0.301** | 0.342 | 0.304 | 0.341 | 0.306 |
| ETTm1 | 0.441 | 0.471 | 0.424 | 0.447 | 0.411 | 0.436 | **0.409** | **0.434** | 0.413 | 0.440 |
| ETTm2 | 0.281 | 0.206 | 0.278 | 0.201 | **0.273** | **0.194** | 0.276 | 0.196 | 0.277 | 0.199 |
| Weather | 0.235 | 0.191 | 0.230 | 0.185 | **0.228** | **0.184** | 0.231 | 0.186 | **0.228** | **0.184** |

without drifting too far from the anchor distribution.

**Forecasting time horizons.** In Table 13 in Appendix D, we report ReAugment's performance under varying prediction horizons (96, 192, 336, 720) in the few-shot setup. The improvements persist under longer horizons, indicating that the learned augmentations do not merely fit the default prediction length but also help the model generalize when the forecasting task becomes more difficult.

### 5.5. Results with Full Access to Training Set

ReAugment can also be applied to standard time series forecasting scenarios, where we have full access to the entire training sets. As shown in Table 10, ReAugment delivers significant performance improvements across multiple datasets in the standard setup, highlighting its strong generalizability beyond the few-shot learning context. Other experimental details, such as the lookback and prediction lengths, are consistent with those in the few-shot setup. Although the benefit is naturally more pronounced under data scarcity, the full-data results show that targeted augmentation can still

refine the training distribution when more observations are available.

## 6. Conclusions and Limitations

In this paper, we proposed ReAugment, a novel reinforcement learning-driven framework for few-shot time series forecasting. Our approach introduces two key innovations: (i) a model-zoo-based diagnostic that identifies overfit-prone anchor points by measuring prediction diversity, and (ii) a closed-loop RL strategy using GRPO to fine-tune a generative augmentor for task-aware data synthesis. Experimental results across multiple benchmarks demonstrate that ReAugment significantly enhances the generalization of various forecasting models.

One unresolved issue in this study is the reliance on multiple pretrained models. The proposed method introduces additional computational overhead due to the need for training VMAE, applying the GRPO algorithm, and performing backtesting across the model zoo.

*Table 10.* **Performance under the standard setup with full training data.** All methods use the iTransformer backbone and the same set of random seeds. Reported values are the mean of three runs. See Appendix Table 17 for the corresponding standard deviations.

| Dataset | Original | | Gaussian | | Convolve | | TimeGAN | | ADA | | ReAugment | |
| | MAE | MSE | MAE | MSE | MAE | MSE | MAE | MSE | MAE | MSE | MAE | MSE |
| --- | --- | --- | --- | --- | --- | --- | --- | --- | --- | --- | --- | --- |
| ETTh1 | 0.405 | 0.387 | 0.407 | 0.392 | 0.416 | 0.399 | 0.409 | 0.390 | 0.407 | 0.391 | **0.397**±0.001 | **0.383**±0.001 |
| ETTh2 | 0.350 | 0.301 | 0.352 | 0.307 | 0.356 | 0.303 | 0.348 | 0.299 | 0.347 | 0.297 | **0.343**±0.002 | **0.292**±0.001 |
| ETTm1 | 0.377 | 0.341 | 0.374 | 0.340 | 0.387 | 0.352 | 0.392 | 0.357 | 0.372 | 0.336 | **0.365**±0.001 | **0.327**±0.001 |
| ETTm2 | 0.272 | 0.186 | 0.272 | 0.187 | 0.275 | 0.188 | 0.279 | 0.190 | 0.273 | 0.188 | **0.262**±0.002 | **0.178**±0.001 |
| Weather | 0.219 | 0.178 | 0.227 | 0.187 | 0.265 | 0.210 | 0.219 | 0.177 | 0.222 | 0.180 | **0.208**±0.001 | **0.171**±0.000 |
| Electricity | 0.239 | 0.148 | 0.243 | 0.150 | 0.264 | 0.170 | 0.276 | 0.183 | 0.241 | 0.149 | **0.233**±0.001 | **0.145**±0.001 |
| Traffic | 0.269 | 0.392 | 0.269 | 0.394 | 0.283 | 0.407 | 0.296 | 0.412 | 0.268 | 0.391 | **0.265**±0.001 | **0.389**±0.001 |
| Exchange | 0.206 | 0.086 | 0.208 | 0.087 | 0.210 | 0.087 | 0.210 | 0.087 | 0.209 | 0.087 | **0.203**±0.001 | **0.085**±0.000 |

## Acknowledgments

This work was supported by the Smart Grid National Science and Technology Major Project (2024ZD0801200), the National Natural Science Foundation of China (62250062), the Shanghai Municipal Science and Technology Major Project (2021SHZDZX0102), the Fundamental Research Funds for the Central Universities, and the Shanghai Jiao Tong University AI for Engineering Initiative (WH410263001/005).

## Impact Statement

In this work, we adhere to the highest ethical standards across all stages of research. No human subjects were involved, and no personal data was used, ensuring compliance with privacy and security protocols. All datasets utilized are publicly available, mitigating concerns related to sensitive information exposure. We acknowledge the potential for forecasting models to generate harmful insights if misapplied; therefore, we encourage careful consideration of the context and application domain when deploying these models.

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

# Appendix

## A. Overall Training Pipeline

We present the overall training pipeline in Algorithm 1.

---

**Algorithm 1** Overall training pipeline

---

1: **Given:** Time series samples from training set $(x^{(i)}, y^{(i)})$
2: **Key problem:** Which samples should be augmented and how to augment them?
3: // Data filtering by model zoo variance
4: Pretrain a forecasting model zoo.
5: Select top $50\%$ high-variance samples as anchor set $\{(x, y)\}$.
6: // Train the VMAE supervised by original data
7: VMAE is parameterized by $\theta_1, \theta_2, \theta_3, \phi$, all optimized during pretraining.
8: // GRPO with model zoo
9: Fix $\theta_2, \theta_3, \phi$ and optimize the prior network $\theta_1$.
10: **while** not converged **do**
11:     Sample a batch of masked inputs $(m, t)$.
12:     For each input, sample a group of latent codes:

$$z^{(i)} \sim \pi_{\theta_1}(z \mid m, t), \quad i = 1, \ldots, G.$$

13:     Decode each $z^{(i)}$ to obtain augmented samples and compute rewards $r^{(i)}$ via model zoo.
14:     Compute group-relative advantages:

$$A^{(i)} = r^{(i)} - \frac{1}{G} \sum_{j=1}^{G} r^{(j)}.$$

15:     Update policy parameters:

$$\theta_1 \leftarrow \theta_1 + \alpha \cdot \frac{1}{G} \sum_{i=1}^{G} A^{(i)} \nabla_{\theta_1} \log \pi_{\theta_1}\left(z^{(i)} \mid m, t\right).$$

16: **end while**
17: // Train the forecasting model
18: Generate augmented data via ReAugment.
19: Train the forecasting model (*e.g.*, iTransformer) with original + augmented data.

---

## B. Dataset Details

Here is a detailed description of the five experiment datasets:

1. ETT consists of two hourly-level datasets (ETTh) and two 15-minute-level datasets (ETTm). Each of them contains 7 factors of electricity transformers, including load and oil temperature from July 2016 to July 2018.

2. Traffic is a collection of road occupancy rates measured by 862 sensors on San Francisco Bay area freeways from January 2015 to December 2016.

3. ECL collects hourly electricity consumption of 321 clients from 2012 to 2014.

4. The Weather dataset includes 21 meteorological indicators, such as air temperature and humidity, recorded 10 minutes from the weather station of the Max Planck Biogeochemistry Institute in 2020.

5. The Exchange dataset records the daily exchange rates of 8 different countries ranging from 1990 to 2016.

For the standard setup, we follow the data processing method of iTransformer, dividing the dataset into training, validation, and test sets, with this partitioning aligned in chronological order.

*Table 11.* **Details of the datasets.** *Features* denotes the number of data variables in each dataset. *Time points* refers to the total number of time points in the dataset. *Partition* indicates the number of time points allocated to each subset in the (train, validation, test) splits.

| | ETTh1 / ETTh2 | ETTm1 / ETTm2 | Traffic |
|---|---|---|---|
| Features | 7 | 7 | 862 |
| Time points (Standard) | 14307 | 57507 | 17451 |
| Time points (Few-shot) | 8443 | 28793 | 8756 |
| Partition (Standard) | (8545, 2881, 2881) | (34465, 11521, 11521) | (12185, 1757, 3509) |
| Partition (Few-shot) | (2681, 2881, 2881) | (5751, 11521, 11521) | (3490, 1757, 3509) |
| | Electricity | Weather | Exchange |
| Features | 321 | 21 | 8 |
| Time points (Standard) | 26211 | 52603 | 7207 |
| Time points (Few-shot) | 13136 | 21071 | 3528 |
| Partition (Standard) | (18317, 2633, 5261) | (36792, 5271, 10540) | (5120, 665, 1422) |
| Partition (Few-shot) | (5242, 2633, 5261) | (5260, 5271, 10540) | (1441, 665, 1422) |

*Table 12.* **Hyperparameters of ReAugment.**

| Notation | Hyperparameter | Description |
|---|---|---|
| $\alpha$ | 0.001 | Learning rate of GRPO |
| $\beta$ | 0.1 | Weight of KL-divergence in the VMAE loss function |
| $L$ | 96 | Time series periods length |
| $\eta$ | 0.01 | Parameters of scaled sigmoid |
| $N$ | 32 | Batch size for VMAE training |

In addition, to simulate a scenario with limited training data, we propose the few-shot setup. Specifically, we reduce the training set size to either $10\%$ or $20\%$ of the full dataset, while keeping the validation and test sets the same as in the standard setup. Notably, the training data used consists of the most distant portion of the time series relative to the test set, to simulate a more challenging time series forecasting task. In Table 11, we provide the number of variables (*i.e.*, the feature dimension at a single time point) in each dataset, the total number of time points, and the number of time points within each set of the train-validation-test partitions for both standard and few-shot setup.

## C. Hyperparameters

In Table 12, we provide the hyperparameter details of VMAE and GRPO. For the encoder and decoder, we adopt the identical hyperparameters as those employed in iTransformer.

## D. Experiments on Prediction Horizons

In the main text, we employ a fixed lookback length of 96 time steps across all datasets and report the prediction results with a fixed horizon of 96 time steps. In this section, we explore the impact of different prediction lengths on the performance of our method. Specifically, we evaluate our model's performance with a prediction horizon of varying lengths, including 96, 192, 336, and 720 time steps.

The experiments aim to provide insights into how the performance of ReAugment scales with longer prediction horizons, and to assess whether the model's ability to generalize across different time scales holds up as the forecasting task becomes more challenging. As shown in Table 13, ReAugment consistently yields performance improvements across different prediction horizons.

## E. Full Results with Multiple Training Seeds

In the main text, for all stochastic data augmentation methods, we conducted experiments using three different random seeds and reported the mean performance. It is worth noting that in all experiments, the source of randomness stems solely from the data augmentation methods, rather than the forecasting models. Accordingly, for stochastic augmentation methods, we applied different random seeds during the augmentation process, while keeping the random seed fixed for the training of the forecasting models. Here, we provide the complete results, including both the mean and standard deviation, for the

*Table 13.* **ReAugment performance under varying prediction horizons** (96, 192, 336, 720) **in the few-shot setup.** All experiments use a fixed input sequence length of 96 time steps.

| Forecast Horizon | 96 | | 192 | | 336 | | 720 | |
|---|---|---|---|---|---|---|---|---|
| | MAE | MSE | MAE | MSE | MAE | MSE | MAE | MSE |
| ETTh1 (Raw) | 0.434 | 0.411 | 0.468 | 0.474 | 0.491 | 0.520 | 0.527 | 0.539 |
| + ReAugment | 0.423 | 0.402 | 0.457 | 0.459 | 0.479 | 0.504 | 0.508 | 0.526 |
| ETTh2 (Raw) | 0.362 | 0.320 | 0.419 | 0.397 | 0.453 | 0.448 | 0.469 | 0.449 |
| + ReAugment | 0.337 | 0.301 | 0.399 | 0.380 | 0.426 | 0.421 | 0.438 | 0.421 |
| ETTm1 (Raw) | 0.440 | 0.470 | 0.471 | 0.513 | 0.480 | 0.585 | 0.533 | 0.638 |
| + ReAugment | 0.411 | 0.436 | 0.437 | 0.475 | 0.444 | 0.548 | 0.501 | 0.593 |
| ETTm2 (Raw) | 0.282 | 0.204 | 0.331 | 0.270 | 0.371 | 0.339 | 0.429 | 0.421 |
| + ReAugment | 0.273 | 0.194 | 0.323 | 0.260 | 0.359 | 0.329 | 0.412 | 0.396 |
| Weather (Raw) | 0.231 | 0.187 | 0.273 | 0.235 | 0.319 | 0.315 | 0.372 | 0.384 |
| + ReAugment | 0.228 | 0.184 | 0.271 | 0.231 | 0.306 | 0.312 | 0.362 | 0.372 |
| Electricity (Raw) | 0.258 | 0.168 | 0.270 | 0.181 | 0.289 | 0.197 | 0.340 | 0.248 |
| + ReAugment | 0.255 | 0.165 | 0.266 | 0.178 | 0.283 | 0.194 | 0.334 | 0.241 |
| Traffic (Raw) | 0.318 | 0.466 | 0.329 | 0.484 | 0.340 | 0.497 | 0.371 | 0.542 |
| + ReAugment | 0.291 | 0.427 | 0.298 | 0.452 | 0.321 | 0.466 | 0.350 | 0.511 |
| Exchange (Raw) | 0.228 | 0.103 | 0.327 | 0.195 | 0.473 | 0.358 | 0.725 | 0.892 |
| + ReAugment | 0.224 | 0.098 | 0.320 | 0.191 | 0.469 | 0.357 | 0.711 | 0.881 |

following experiments:

- The impact of ReAugment on different forecasting models under few-shot learning in Table 14;

- The few-shot forecasting performance using different data augmentation methods in Table 15;

- The few-shot performance of different data augmentation methods evaluated using our new metrics, with mean and standard deviation reported in Table 16.

- The performance of ReAugment under the standard setup with the full training set in Table 17.

## F. Complete Results for Preliminary Findings

Table 18 provides the full cross-dataset results supporting the preliminary finding in Section 3.2 that sample-wise variance correlates with forecasting difficulty. For each dataset, we report forecasting performance when training on the top-variance and bottom-variance halves of the data, corresponding to the analysis introduced in the main text.

*Table 14.* **Impact of ReAugment on different forecasting models in the few-shot learning setup.** For the raw data baseline, no standard deviation is reported since the forecasting model is trained with a fixed random seed. We report both the mean and standard deviation for ReAugment.

| Training Data | iTransformer | | PatchTST | | DLinear | |
| --- | --- | --- | --- | --- | --- | --- |
| | MAE | MSE | MAE | MSE | MAE | MSE |
| ETTh1 (Raw) | 0.434 | 0.411 | 0.458 | 0.446 | 0.435 | 0.408 |
| + ReAugment | 0.423±0.002 | 0.402±0.001 | 0.438±0.002 | 0.430±0.002 | 0.420±0.002 | 0.389±0.001 |
| ETTh2 (Raw) | 0.362 | 0.320 | 0.367 | 0.321 | 0.402 | 0.356 |
| + ReAugment | 0.337±0.001 | 0.301±0.001 | 0.347±0.001 | 0.305±0.001 | 0.368±0.001 | 0.332±0.001 |
| ETTm1 (Raw) | 0.440 | 0.470 | 0.428 | 0.457 | 0.442 | 0.471 |
| + ReAugment | 0.411±0.001 | 0.436±0.002 | 0.404±0.002 | 0.432±0.002 | 0.430±0.001 | 0.460±0.002 |
| ETTm2 (Raw) | 0.282 | 0.204 | 0.276 | 0.199 | 0.303 | 0.219 |
| + ReAugment | 0.273±0.001 | 0.194±0.001 | 0.267±0.001 | 0.194±0.001 | 0.296±0.000 | 0.217±0.000 |
| Weather (Raw) | 0.231 | 0.187 | 0.232 | 0.189 | 0.277 | 0.212 |
| + ReAugment | 0.228±0.001 | 0.184±0.001 | 0.225±0.000 | 0.187±0.000 | 0.274±0.001 | 0.211±0.000 |
| Electricity (Raw) | 0.258 | 0.168 | 0.295 | 0.200 | 0.307 | 0.215 |
| + ReAugment | 0.255±0.001 | 0.165±0.000 | 0.276±0.001 | 0.186±0.001 | 0.306±0.001 | 0.214±0.001 |
| Traffic (Raw) | 0.318 | 0.466 | 0.327 | 0.541 | 0.452 | 0.724 |
| + ReAugment | 0.291±0.001 | 0.427±0.001 | 0.315±0.002 | 0.520±0.002 | 0.404±0.001 | 0.664±0.002 |
| Exchange (Raw) | 0.228 | 0.103 | 0.226 | 0.103 | 0.226 | 0.104 |
| + ReAugment | 0.224±0.000 | 0.098±0.000 | 0.221±0.000 | 0.099±0.000 | 0.225±0.000 | 0.100±0.000 |

*Table 15.* **Few-shot forecasting results using various augmentation methods.** We use iTransformer as the forecasting model and report results across three random seeds.

| Dataset | Original | | Gaussian | | Convolve | |
| --- | --- | --- | --- | --- | --- | --- |
| | MAE | MSE | MAE | MSE | MAE | MSE |
| ETTh1 | 0.434 | 0.411 | 0.437±0.002 | 0.416±0.002 | 0.441±0.003 | 0.417±0.002 |
| ETTh2 | 0.362 | 0.320 | 0.365±0.001 | 0.321±0.001 | 0.364±0.001 | 0.323±0.001 |
| ETTm1 | 0.440 | 0.470 | 0.438±0.003 | 0.469±0.002 | 0.426±0.002 | 0.479±0.003 |
| ETTm2 | 0.282 | 0.204 | 0.283±0.001 | 0.204±0.001 | 0.286±0.001 | 0.207±0.001 |
| Weather | 0.231 | 0.187 | 0.240±0.001 | 0.196±0.000 | 0.253±0.001 | 0.204±0.001 |
| Electricity | 0.258 | 0.168 | 0.263±0.001 | 0.170±0.001 | 0.262±0.001 | 0.170±0.001 |
| Traffic | 0.318 | 0.466 | 0.319±0.001 | 0.467±0.002 | 0.320±0.001 | 0.463±0.001 |
| Exchange | 0.228 | 0.103 | 0.229±0.000 | 0.104±0.000 | 0.226±0.001 | 0.099±0.000 |

| Dataset | TimeGAN | | ADA | | ReAugment | |
| --- | --- | --- | --- | --- | --- | --- |
| ETTh1 | 0.444±0.002 | 0.419±0.001 | 0.435±0.001 | 0.413±0.001 | **0.423**±0.002 | **0.402**±0.001 |
| ETTh2 | 0.366±0.001 | 0.327±0.001 | 0.368±0.000 | 0.331±0.000 | **0.337**±0.001 | **0.301**±0.001 |
| ETTm1 | 0.430±0.003 | 0.483±0.003 | 0.429±0.001 | 0.484±0.001 | **0.411**±0.001 | **0.436**±0.002 |
| ETTm2 | 0.285±0.002 | 0.206±0.001 | 0.284±0.001 | 0.207±0.001 | **0.273**±0.001 | **0.194**±0.001 |
| Weather | 0.239±0.001 | 0.191±0.001 | 0.246±0.000 | 0.198±0.000 | **0.228**±0.001 | **0.184**±0.001 |
| Electricity | 0.267±0.001 | 0.177±0.001 | 0.265±0.001 | 0.171±0.000 | **0.255**±0.001 | **0.165**±0.000 |
| Traffic | 0.315±0.002 | 0.449±0.003 | 0.318±0.001 | 0.456±0.001 | **0.291**±0.001 | **0.427**±0.001 |
| Exchange | 0.226±0.001 | 0.100±0.001 | 0.235±0.000 | 0.116±0.000 | **0.224**±0.000 | **0.098**±0.000 |

*Table 16.* **Few-shot time series forecasting results in our new metrics.** To assess the statistical robustness of these new metrics, we report their mean and standard deviation over three runs.

| Dataset | Metric | Gaussian | Convolve | TimeGAN | ADA | ReAugment |
|---|---|---|---|---|---|---|
| ETTh1 | $\mathcal{F}_{\textbf{MAE}}$ | -10.3%±6.8% | -24.1%±9.3% | -34.5%±6.4% | -3.4%±3.1% | **37.9%±4.7%** |
| | $\mathcal{F}_{\textbf{MSE}}$ | -20.8%±7.3% | -25.0%±6.9% | -33.3%±4.5% | -8.3%±3.9% | **37.5%±3.3%** |
| ETTh2 | $\mathcal{F}_{\textbf{MAE}}$ | -25.0%±8.5% | -16.7%±7.7% | -33.3%±8.2% | -50.0%±3.1% | **208.3%±6.9%** |
| | $\mathcal{F}_{\textbf{MSE}}$ | -5.3%±4.3% | -15.8%±3.9% | -36.8%±6.8% | -57.9%±2.4% | **100.0%±6.2%** |
| ETTm1 | $\mathcal{F}_{\textbf{MAE}}$ | 3.2%±4.6% | 22.2%±3.1% | 15.9%±4.3% | 17.5%±1.8% | **46.0%±1.9%** |
| | $\mathcal{F}_{\textbf{MSE}}$ | 0.8%±1.5% | -7.0%±2.3% | -10.1%±2.1% | -10.8%±0.8% | **26.4%±2.7%** |
| ETTm2 | $\mathcal{F}_{\textbf{MAE}}$ | -10.0%±8.9% | -40.0%±13.1% | -30.0%±18.8% | -20.0%±8.9% | **90.0%±11.6%** |
| | $\mathcal{F}_{\textbf{MSE}}$ | 0.0%±4.9% | -16.7%±5.7% | -11.1%±6.2% | -16.7%±4.4% | **55.6%±5.1%** |
| Weather | $\mathcal{F}_{\textbf{MAE}}$ | -75.0%±10.2% | -183.3%±9.1% | -66.7%±8.3% | -125%±2.9% | **25.0%±3.1%** |
| | $\mathcal{F}_{\textbf{MSE}}$ | -100.0%±2.4% | -188.9%±14.2% | -44.4%±8.9% | -122.2%±4.1% | **33.3%±2.2%** |
| Electricity | $\mathcal{F}_{\textbf{MAE}}$ | -26.3%±4.7% | -21.1%±6.3% | -47.4%±5.8% | -36.8%±6.1% | **15.8%±4.4%** |
| | $\mathcal{F}_{\textbf{MSE}}$ | -10.0%±6.0% | -10.0%±4.4% | -45.0%±4.6% | -15.0%±1.3% | **15.0%±2.1%** |
| Traffic | $\mathcal{F}_{\textbf{MAE}}$ | -2.0%±1.8% | -4.1%±2.2% | 6.1%±4.4% | 0.0%±1.4% | **55.1%±2.2%** |
| | $\mathcal{F}_{\textbf{MSE}}$ | -1.4%±3.0% | 4.1%±1.2% | 23.0%±4.3% | 13.5%±1.7% | **52.7%±2.0%** |
| Exchange | $\mathcal{F}_{\textbf{MAE}}$ | -4.5%±0.9% | 9.1%±4.8% | 9.1%±3.8% | -31.8%±2.5% | **18.2%±1.5%** |
| | $\mathcal{F}_{\textbf{MSE}}$ | -5.9%±1.4% | 23.5%±2.2% | 17.6%±5.9% | -76.5%±2.1% | **29.4%±1.7%** |

*Table 17.* **Comparison under standard setup with full training set.** We use iTransformer as the forecasting model and report the mean and standard deviation across three random seeds.

| Dataset | Original | | Gaussian | | Convolve | |
|---|---|---|---|---|---|---|
| | MAE | MSE | MAE | MSE | MAE | MSE |
| ETTh1 | 0.405 | 0.387 | 0.407±0.002 | 0.392±0.001 | 0.416±0.003 | 0.399±0.002 |
| ETTh2 | 0.350 | 0.301 | 0.352±0.001 | 0.307±0.001 | 0.356±0.001 | 0.303±0.001 |
| ETTm1 | 0.377 | 0.341 | 0.374±0.002 | 0.340±0.002 | 0.387±0.002 | 0.352±0.002 |
| ETTm2 | 0.272 | 0.186 | 0.272±0.001 | 0.187±0.000 | 0.275±0.001 | 0.188±0.001 |
| Weather | 0.219 | 0.178 | 0.227±0.001 | 0.187±0.001 | 0.265±0.001 | 0.210±0.001 |
| Electricity | 0.239 | 0.148 | 0.243±0.001 | 0.150±0.000 | 0.264±0.001 | 0.170±0.001 |
| Traffic | 0.269 | 0.392 | 0.269±0.001 | 0.394±0.002 | 0.283±0.002 | 0.407±0.002 |
| Exchange | 0.206 | 0.086 | 0.208±0.000 | 0.087±0.000 | 0.210±0.001 | 0.087±0.000 |

| Dataset | TimeGAN | | ADA | | ReAugment | |
|---|---|---|---|---|---|---|
| ETTh1 | 0.409±0.002 | 0.390±0.002 | 0.407±0.001 | 0.391±0.001 | **0.397±0.001** | **0.383±0.001** |
| ETTh2 | 0.348±0.001 | 0.299±0.001 | 0.347±0.000 | 0.297±0.000 | **0.343±0.002** | **0.292±0.001** |
| ETTm1 | 0.392±0.002 | 0.357±0.003 | 0.372±0.001 | 0.336±0.002 | **0.365±0.001** | **0.327±0.001** |
| ETTm2 | 0.279±0.001 | 0.190±0.001 | 0.273±0.000 | 0.188±0.000 | **0.262±0.002** | **0.178±0.001** |
| Weather | 0.219±0.001 | 0.177±0.001 | 0.222±0.000 | 0.180±0.000 | **0.208±0.001** | **0.171±0.000** |
| Electricity | 0.276±0.002 | 0.183±0.001 | 0.241±0.001 | 0.149±0.001 | **0.233±0.001** | **0.145±0.001** |
| Traffic | 0.296±0.002 | 0.412±0.003 | 0.268±0.002 | 0.391±0.002 | **0.265±0.001** | **0.389±0.001** |
| Exchange | 0.210±0.001 | 0.087±0.000 | 0.209±0.000 | 0.087±0.000 | **0.203±0.001** | **0.085±0.000** |

*Table 18.* **Forecasting results on high-variance vs. low-variance subsets across all datasets.**

| Subset | ETTh1 | | ETTh2 | | ETTm1 | | ETTm2 | | Weather | | Electricity | | Traffic | | Exchange | |
|---|---|---|---|---|---|---|---|---|---|---|---|---|---|---|---|---|
| | MAE | MSE | MAE | MSE | MAE | MSE | MAE | MSE | MAE | MSE | MAE | MSE | MAE | MSE | MAE | MSE |
| Top 50% variance | 0.418 | 0.399 | 0.357 | 0.305 | 0.402 | 0.359 | 0.272 | 0.187 | 0.224 | 0.185 | 0.256 | 0.161 | 0.289 | 0.413 | 0.210 | 0.087 |
| Bottom 50% variance | 0.403 | 0.386 | 0.344 | 0.297 | 0.362 | 0.329 | 0.258 | 0.177 | 0.217 | 0.176 | 0.250 | 0.154 | 0.292 | 0.415 | 0.205 | 0.085 |
| Full training set | 0.405 | 0.387 | 0.350 | 0.301 | 0.377 | 0.341 | 0.272 | 0.186 | 0.219 | 0.178 | 0.239 | 0.148 | 0.269 | 0.392 | 0.206 | 0.086 |

