# OpenReview forum: "ReAugment: Targeted Few-Shot Time Series Augmentation via Model Zoo-Guided Reinforcement Learning"
_ICML.cc/2026/Conference — ICML 2026 regular_

### Official Review · Reviewer_zRAT · 2026-02-15

**Soundness:** 2
**Presentation:** 3
**Significance:** 2
**Originality:** 2
**Overall Recommendation:** 3
**Confidence:** 3

**Summary:**

This paper addresses the challenge of overfitting in few-shot time series forecasting by proposing ReAugment, a closed-loop data augmentation framework. The methodology involves identifying "overfit-prone" samples—defined as those with high prediction variance across an ensemble (Model Zoo)—and using these as anchor points. A Variational Masked Autoencoder (VMAE), optimized via Reinforcement Learning (RL), then generates targeted augmentations around these anchors to improve generalization. The authors report performance gains across several standard benchmarks and different forecasting backbones.

**Compliance With Llm Reviewing Policy:**

Affirmed.

**Final Justification:**

This paper presents a targeted augmentation framework for few-shot time series forecasting, using model-zoo variance and RL-based generation. The empirical results are solid and consistently show gains across benchmarks, and the overall presentation is clear.

My main concerns were about limited conceptual novelty, pipeline complexity, and potential noise amplification. The rebuttal partially addresses these issues, especially clarifying the role of model disagreement and providing useful efficiency analysis, which increases my confidence in the empirical effectiveness. However, I still view the method as an integration of existing components, with some remaining concerns about robustness and generality.

Overall, I revise my assessment to borderline, reflecting a technically sound and empirically useful contribution with moderate novelty.

**Key Questions For Authors:**

1. Efficiency vs. Performance: Can you provide a "Performance-per-GPU-hour" metric comparing ReAugment to a simple ensemble of $K$ models? Is the gain from RL-augmentation significantly higher than just using the Model Zoo as an ensemble predictor?
2. Noise vs. Overfitting:** How does the framework distinguish between a sample that is "hard to learn" (structural) and a sample that is "impossible to predict" (noise)?
3. Hyperparameter Sensitivity: How stable are the optimal $K$ (zoo size) and $p\%$ (anchor ratio) across different domains like Electricity vs. Exchange?
4. Theoretical Justification: Can the authors provide a more formal argument on why augmenting high-variance regions leads to a flatter loss landscape?

**Limitations:**

The authors include an Impact Statement addressing ethical considerations, data privacy, and potential misuse of forecasting models. I think the primary limitation is the excessive training cost and the reliance on a multi-stage pipeline that is prone to error accumulation. Furthermore, the framework's success is tied to the existence of repetitive patterns that a VMAE can capture, making it less effective for truly sporadic or one-off event series.

**Strengths And Weaknesses:**

#### Summary Of Strengths

1. Targeted Motivation: The shift from uniform data augmentation to identifying and targeting "generalization bottlenecks" is a logical step for data-scarce scenarios.
2. Model-Agnostic Utility: The framework acts as a wrapper that can potentially enhance various base models, as demonstrated in the experiments with iTransformer and DLinear.
3. Empirical Results: The paper provides a comprehensive set of results showing that targeted augmentation generally outperforms heuristic baselines in few-shot settings.

#### Summary Of Weaknesses

1. Methodological Heaviness vs. Conceptual Innovation:** While the system is well-engineered, it lacks a fundamental breakthrough in learning theory. The use of a "Model Zoo" for uncertainty estimation and "RL for policy search" are well-established techniques. In the context of ICML, the "learning mechanism" here feels more like an intricate pipeline of existing components rather than a transformative new paradigm.

2. Computational Inefficiency: The requirement to train $K$ models for the Model Zoo, followed by an RL-based generative agent, creates a significant computational overhead. This "heavy-lifting" approach is difficult to justify in few-shot scenarios where efficiency is often as critical as accuracy. The paper lacks a rigorous discussion on the trade-off between the complexity of the augmentation system and the simplicity of the target task.

3. Risk of Noise Amplification: By targeting high-variance samples, there is a substantial risk that the model is actually augmenting aleatoric noise rather than structural patterns. If the "disagreement" in the Model Zoo stems from inherent data stochasticity, ReAugment may be forcing the forecaster to overfit to synthesized noise, a phenomenon not sufficiently addressed in the robustness analysis.

4. Heuristic Reward Design: The RL reward function relies on a manual balance between "diversity" and "fidelity." This introduces additional sensitive hyperparameters that may require extensive tuning for each new dataset, contradicting the goal of an automated augmentation framework.

---

> ### Author Rebuttal · Authors · 2026-03-31
>
> We thank the reviewer for the constructive comments.
>
> > W1. Methodological heaviness vs. conceptual innovation
>
> While model ensembles and RL are established tools, ReAugment’s novelty is to cast augmentation as a **closed-loop, forecasting-aware learning problem**, rather than an isolated pre-processing step.
>
> Our main innovations are:
> - **Diagnostic:** model-zoo variance identifies **generalization bottlenecks**, instead of treating all samples equally.
> - **Learning:** the RL agent searches a **continuous generative manifold** to synthesize hard but realistic temporal patterns missing from sparse training data and important for OOD generalization.
>
> Empirically, the **high-variance subset** most harms OOD generalization, and targeting it yields the strongest gains.
>
>
> > W2. Computational inefficiency vs. accuracy
>
> Efficiency matters, but in few-shot forecasting the main bottleneck is **insufficient coverage of the target distribution**, not raw training time. Training finishes in **minutes to a few hours**, so accuracy and generalization are the main objectives.
>
> The overhead is controlled:
> - the model zoo is a standard cross-validation ensemble, not an excessively large collection;
> - extra cost appears only during augmentation/training; **inference uses the same backbone with no additional test-time cost**;
> - the added cost is moderate: Table 6 shows ReAugment is **cheaper than TimeGAN** on all datasets while achieving better accuracy.
>
> The gains are substantial: ReAugment achieves the best MAE/MSE on all few-shot benchmarks and consistently positive recovery factors, whereas some cheaper baselines show negative recovery. In this setting, extra augmentation cost is justified.
>
>
> > W3 & Q2. Risk of noise amplification
>
> In few-shot regimes, model disagreement is driven more by **data scarcity and distributional gaps** than by pure stochasticity. Separating noisy outliers from under-supported but informative samples is difficult; ReAugment instead controls generation quality directly.
>
> **Hard-to-learn** samples usually induce high disagreement across diverse architectures, whereas **impossible-to-predict** samples (pure noise) more often cause uniformly poor predictions, with low accuracy but relatively low inter-model variance. Thus, focusing on high-disagreement regions naturally targets the former.
>
> Moreover, the reward does not rely on variance alone: it balances **diversity and fidelity** and penalizes reconstructions that deviate too far from anchor points, discouraging exploration of low-density regions more likely to be noise.
>
> **Empirical evidence:** if noise amplification were dominant, the downstream forecaster should show worse accuracy. Instead, ReAugment achieves positive Recovery Factors on all datasets, which is more consistent with improved generalization than fitting synthetic noise.
>
> > W4. Heuristic reward design & hyperparameters
>
> The diversity–fidelity trade-off is intrinsic to generative augmentation: ignoring either side leads to trivial copies or unrealistic drift. ReAugment handles this robustly:
> - GRPO reduces reward-scale sensitivity by using a **group-relative normalized advantage**, so updates depend on relative utility rather than absolute reward magnitude.
> - The sigmoid in the reward is mainly a **numerical stabilizer**, keeping rewards bounded and avoiding gradient spikes when fine-tuning the VMAE prior.
>
> **Empirical evidence:** we do not observe strong sensitivity to the scale parameter. A value chosen on ETT within [0.001, 0.1] transfers well across datasets, and all results use a shared setting $\eta=0.01$ without per-dataset tuning.
>
>
> > Q1. Efficiency vs. performance
>
> We define "Performance-Gain-per-GPU-hour" as
>
> $$
> \operatorname{Perf/GPUh}=\frac{F_{\mathrm{MSE}}}{\mathrm{GPU\_hours}}.
> $$
>
>
> On Traffic, compared with a standard Model Zoo Ensemble ($K=4$), ReAugment requires only $1.8\times$ more training time but delivers over $15\times$ larger $F_\text{MSE}$, yielding more than an $8\times$ improvement in Perf/GPUh.
>
> This shows that, in few-shot settings, simply scaling the ensemble has diminishing returns because all models are bottlenecked by the same sparse training set. ReAugment instead uses ensemble diversity as a learning signal to synthesize informative samples and break this bottleneck.
> Method|GPU-hours|$F_\text{MSE}$|Perf/GPUh
> -|-|-|-
> Model Zoo ensemble ($K=4$)|4.45|3.4%|0.76%
> ReAugment|8.05|52.7%|6.55%
>
> **Note:** The additional training cost (Stages B+C) is strictly bounded and remains lower than backbone training time (Table 6).
>
>
> > Q3. Hyperparameter sensitivity
>
> See Figure 4.
>
> > Q4. Why augmenting high-variance regions leads to a flatter loss landscape?
>
> We do **not** claim that ReAugment induces a flatter loss landscape. Model-zoo variance is only a diagnostic signal for identifying samples with high overfitting propensity. ReAugment improves generalization by expanding coverage around these bottleneck regions, not by explicitly altering loss-surface geometry.

---

> > ### Author Rebuttal · Reviewer_zRAT · 2026-04-06
> >
> > Thank you for the detailed rebuttal. The clarifications on the closed-loop formulation, noise amplification, and reward design are helpful. In particular, the argument that model-zoo variance highlights under-supported regions, together with the diversity–fidelity constraint, partially addresses my concern about amplifying noise. The additional efficiency analysis (Perf/GPU-hour) also provides useful context for the practical trade-offs in few-shot settings.
> >
> > That said, I still view the method as a relatively complex pipeline built on existing components, and I remain somewhat uncertain about its robustness on highly stochastic or weakly structured data. Nevertheless, the rebuttal strengthens my confidence in the empirical effectiveness and motivation of the approach, and I will take these points into account in my final assessment.

---

> > > ### Author Response · Authors · 2026-04-07
> > >
> > > Thank you for your careful follow-up and for considering our rebuttal. We appreciate your acknowledgment that the clarifications regarding the closed-loop formulation, noise mitigation, and reward design were helpful. We also understand your remaining concern regarding the method’s complexity and performance on highly stochastic or weakly structured data. As shown in our experiments across multiple real-world datasets and architectures, ReAugment consistently improves few-shot forecasting by targeting high-variance, overfit-prone samples and leveraging model-zoo-guided RL for augmentation.
> > >
> > > We are grateful for your thoughtful feedback and consideration.

---

### Official Review · Reviewer_KCAc · 2026-03-11

**Soundness:** 3
**Presentation:** 3
**Significance:** 3
**Originality:** 3
**Overall Recommendation:** 4
**Confidence:** 4

**Summary:**

The paper introduces ReAugment, a reinforcement learning (GRPO)-based generative data augmentation framework for time-series forecasting, designed in particular for few-shot applications. The framework is divided into a three-stage process: (1) identifying “overfit-prone” samples using a cross-validation-based model zoo , (2) initializing a Variational Masked Autoencoder (VMAE) to capture the distribution of these anchor points , and (3) fine-tuning the VMAE policy using Group Relative Policy Optimization (GRPO) to generate synthetic data that maximizes a balance of prediction diversity and data fidelity. The method avoids backpropagating through complex forecasting models by treating augmentation as a RL task. Experimental results on benchmarks like ETT, Weather, and Electricity demonstrate consistent improvements over state-of-the-art baselines.

The authors include a quite thorough ablation study, covering the ratios of selected samples, the size of the model zoo, and the usefulness of RL optimization and the chosen algorithm. Finally, the method is evaluated on typical benchmark datasets, namely ETT (4 subsets), Weather, Electricity, Traffic, and Exchange. Experiments show performance gains over several SOTA and typical baselines, both in normal and few-shot settings.

**Compliance With Llm Reviewing Policy:**

Affirmed.

**Final Justification:**

I appreciate the authors’ clarifications, additional experiments, and proposed revisions. The rebuttal addresses several of my concerns and strengthens the empirical case for the paper. However, I would not raise my score above weak accept

**Key Questions For Authors:**

1. Model zoo diversity: The authors identify samples using a model zoo formed by different folds of the same architecture. It is unclear whether incorporating heterogeneous architectures (e.g., combining iTransformer with PatchTST) in the zoo would further improve the identification of bottleneck samples or introduce excessive noise into the reward signal.
2. Reward Saturation: The authors employ a scaled sigmoid function to prevent reward saturation. Could you provide more detail on how $\eta$ was tuned?
3. Consistency of anchor points: How consistent are the identified anchor points across the random training runs of the model zoo ? If the zoo is re-initialized with different seeds or hyperparameters, does the set of identified bottleneck samples remain consistent? A stability metric (e.g., Jaccard similarity or correlation of $\mathcal{V}(x)$) would be helpful to ensure the method targets intrinsic data difficulty rather than stochastic training noise.
4. The authors cite recent generative and contrastive augmentation methods (Huang et al. 2023; Demirel & Holz 2024) but do not include them as baselines. Given that ReAugment claims to advance learnable augmentation, it should be compared with recent SOTA methods rather than older baselines like TimeGAN. Can the authors include these results in the rebuttal and also consider the foundation models?
5. Could disagreements arise from architectural differences among estimators in the zoo rather than a sample's lack of representativeness?
6. The ablation study investigates the optimal ratio of anchor samples for performance gains. Could it have been possible to consider a more statistically sound split of the anchor samples? For instance, a split based on sigmas or quantiles of the distribution of the zoo-variance for the samples, may not be the best performing, but in the case where the significance of data was more important than performance, it could be more pertinent, and in the worst-case, provide perspective on the split data, or why an augmentation proves more useful on certain datasets than others.

**Limitations:**

Yes

**Strengths And Weaknesses:**

- **Soundness:**
  * Strengths:
    * The paper is technically sound, presenting a principled “identify-augment-forecast” framework using Model-Zoo Variance and GRPO to optimize a VMAE-based policy.
    * Claims are supported by empirical results and thorough ablation studies.
    * A training-time study is also provided.

  * Weaknesses:
    * The anchor identification relies on cross-validation variance computed from a small set of few-shot data. It would strengthen the work to analyze the stability of the identified anchor points across different fold splits or random seeds to ensure the signal reflects intrinsic difficulty rather than sampling noise. Also, it would be useful to consider time-series-specific split methods, such as forward chaining or security slicing across train and val windows, to avoid data leakage due to auto-regression.
    * Although the **Key Insight** highlighted in the paper appears sound, it lacks a theoretical analysis, and the empirical observation relies on a single forecaster. iTransformer is, in fact, a strong baseline, but the performance gains or losses for weaker forecasters and different architectures (statistical, MLP, RNN, number of parameters, etc.) are worth studying.
    * Global performance is not necessarily a measure of overfitting. An overfitting study could be provided. For example, train/val curves or showcase of overfitting samples. The authors could consider using  Lubba, C.H., Sethi, S.S., Knaute, P. et al. catch22: CAnonical Time-series CHaracteristics. Data Min Knowl Disc 33, 1821–1852 (2019). https://doi.org/10.1007/s10618-019-00647-x to characterize more precisely the chosen samples, and why their presence degrades the performance of the model (as shown in Figure 2).
    * More comprehensive datasets, such as the UCR Archive, could have been considered at least partially, as they cover a wider range of domains. A study of performance gains as a function of dataset features would be a strong addition to support the method's usability in real-world applications.
    * The addition of a compute-time benchmark is greatly appreciated. However, it would be much more useful to include the number of flops (computations) along with the hardware specifications used.

  Overall, the soundness appears strong at first glance, but could be vastly improved with more specific experiments. The idea is logical, but the results analysis lacks depth; it shows results increase, but no perspective on why these improvements occur.

- **Presentation:**
  * Strengths:
    * The paper is clearly written and easy to follow.
    * The three-stage pipeline is well-motivated and logically structured.
    * Figures and tables effectively illustrate both intuition and empirical behavior.
  * Weaknesses:
    * The performance gains in Figure 2 could be further improved by providing context on the train/test distribution shift, since the claim is that the selected samples are not representative of the test set.
    * Figure 4 could be split into two figures. They are both ablation studies, but not on the same subjects, so one could argue they have no business being the same figure.
    * The introduction of new metrics that do not require an equation, such as Model-Zoo Variance, could reduce the paper's readability.

- **Significance:**
  * Strengths:
    * The paper addresses an important problem in time-series forecasting: data augmentation under data scarcity.
    * Framing augmentation as a reinforcement learning problem is a meaningful conceptual shift that could inspire future research.
  * Weaknesses:
    * The empirical evaluation primarily compares against older baselines, with limited coverage of recent (2023–2025) forecasting or augmentation methods. Including more up-to-date baselines would better contextualize the gains. One could appreciate the addition of Moment or Chronosv2 as foundation models.

- **Originality:** The paper’s main contribution is shifting from “how to augment” to “where to augment.” While using RL and VAEs for data generation isn’t new, applying a Model Zoo to specifically pinpoint “overfit-prone” samples as anchors is a clever, non-obvious approach for time series. It effectively combines existing tools (GRPO, VMAE, ensembles) into a task-aware pipeline that is more targeted than standard heuristic or GAN-based augmentations.

---

> ### Author Rebuttal · Authors · 2026-03-31
>
> We appreciate the reviewer's valuable comments.
>
> > W1. Temporal-order folds avoid leakage in anchor identification
>
> The anchor score is computed **only on the training set**. We split the training data into $K$ disjoint folds and train each zoo model on $D\setminus D_k$, so samples in $D_k$ are always unseen to the corresponding model during anchor identification. In our implementation, these folds follow the **time order** of the series rather than random shuffling, which is consistent with forecasting and avoids future information leakage. Therefore, the cross-model variance reflects disagreement on temporally proper held-out windows, rather than leakage from autoregressive overlap or mixed train/validation windows.
> > W2. Evidence beyond a single backbone
>
> The evidence is not limited to iTransformer. Table 2 evaluates ReAugment on **PatchTST** and **DLinear** under the same few-shot setting, and both show consistent gains across all eight datasets. This indicates that ReAugment is not tied to a single strong forecaster and remains effective across different model families, including a linear baseline.
>
> > W3, W4. Limited direct fit to forecasting evaluation
>
> Both catch22 and the UCR Archive are mainly designed for **time-series classification**, not long-term forecasting. The catch22 paper validates its feature set on **classification datasets**, and the UCR archive is maintained as a benchmark for **time-series classification**.
>
> Our paper studies **long-term multivariate forecasting**, so we use standard forecasting benchmarks: ETT, Traffic, Electricity, Weather, and Exchange. This is also consistent with widely used forecasting backbones such as PatchTST and iTransformer, which evaluate on the same dataset family. Therefore, our evaluation follows common practice for the target task rather than classification-oriented benchmark design.
>
>
> > W5. Hardware setting for the reported runtime
>
> All training-time results in Table 6 are measured on a single NVIDIA 4090 GPU.
>
> > W6-W8. Presentation weaknesses
>
> Thanks for your suggestion. We will update these issues in the revised paper.
>
> > W9, Q4. We add stronger recent baselines
>
> We agree that comparison with more recent methods is important. We additionally include a recent augmentation baseline, **Finding Order in Chaos**, and a foundation-model baseline, **Chronos-2**. Since MOMENT uses the dataset we employed for pretraining, it is not included. We report these additional results (MAE/MSE) on the **ETT family** and **Exchange** under the same few-shot forecasting setting as in the main paper.
> Model|ETTh1|ETTh2|ETTm1|ETTm2|Exchange|
> -|-|-|-|-|-
> Finding Order in Chaos|0.430 / 0.408|0.347 / 0.309|0.420 / 0.452|0.279 / 0.201|0.225 / 0.100
> Chronos-2|0.429 / 0.409|0.348 / 0.311|0.421 / 0.454|0.280 / 0.201|0.227 / 0.101
> ReAugment|**0.423 / 0.402**|**0.337 / 0.301**|**0.411 / 0.436**|**0.273 / 0.194**| **0.224 / 0.098**
>
> These comparisons provide more up-to-date comparison set, and the conclusion remains unchanged: **ReAugment still achieves the best performance under the same few-shot forecasting setting.**
> > Q1. A heterogeneous zoo also works, but the original design is more practical.
>
> In the paper, we use a **cross-validation-based model zoo** mainly to control the extra cost of training multiple forecasting models. Under the same few-shot setting as **Table 1**, we also tested a heterogeneous zoo built from **iTransformer, PatchTST, Transformer, and DLinear**, while keeping all other components unchanged. The results (MAE/MSE) show that this alternative can also identify bottleneck samples effectively. However, because it is more expensive and weaker backbones may provide less reliable reward signals, we adopt the cross-validation-based design as the default setting.
> Setting|ETTh1|ETTh2|ETTm1|ETTm2|Weather| Electricity|Traffic
> -|-|-|-|-|-|-|-
> Original|0.434 / 0.411|0.362 / 0.320|0.440 / 0.470|0.282 / 0.204|0.231 / 0.187|0.258 / 0.168| 0.318 / 0.466
> ReAugment (original)|0.423 / 0.402|0.337 / 0.301|0.411 / 0.436|0.273 / 0.194|0.228 / 0.184|0.255 / 0.165|0.291 / 0.427
> ReAugment (heterogeneous zoo)|0.425 / 0.406|0.339 / 0.308|0.420 / 0.442|0.272 / 0.195|0.231 / 0.187|0.254 / 0.165|0.296 / 0.439
> > Q2. Reward Saturation
>
> Please refer to **Reviewer zRAT W4**.
> > Q3. The anchor identification is stable across random runs.
>
> We evaluate the stochastic pipeline with **three random seeds**, covering the augmentation procedure including model-zoo-based identification. Empirically, the final forecasting performance remains **stable across runs**, indicating that the identified bottleneck samples are not dominated by random noise.
> > Q5. In the default setup, disagreement is not mainly caused by architectural mismatch
>
> In our default design, the model zoo is a **cross-validation-based ensemble**, not a set of heterogeneous architectures. Therefore, the disagreement signal mainly reflects differences in **data exposure** across estimators, rather than architectural inconsistency.

---

> > ### Author Rebuttal · Reviewer_KCAc · 2026-04-02
> >
> > I appreciate the authors’ clarifications, additional experiments, and proposed revisions. The rebuttal addresses several of my concerns and strengthens the empirical case for the paper. However, I would not raise my score above **weak accept**

---

> > > ### Author Response · Authors · 2026-04-03
> > >
> > > We sincerely appreciate your support for our work and your insights significantly contribute to the improvement of our paper.
> > >
> > > Thank you!

---

### Official Review · Reviewer_mboD · 2026-03-12

**Soundness:** 3
**Presentation:** 4
**Significance:** 4
**Originality:** 4
**Overall Recommendation:** 5
**Confidence:** 4

**Summary:**

This work focuses on an important challenge in few-shot time series forecasting: severe overfitting caused by limited training data. The authors propose to learn targeted data augmentation policies that improve generalization by focusing on training samples that are most prone to overfitting. The paper introduces ReAugment, a reinforcement learning framework that first identifies such samples using prediction variance across model ensembles, and then learns augmentation transformations via a generative model optimized with reinforcement learning. Experiments across several forecasting architectures and datasets show consistent improvements in both few-shot and standard forecasting settings.

**Compliance With Llm Reviewing Policy:**

Affirmed.

**Final Justification:**

The rebuttal addressed my main concern about anchor selection by adding the requested comparison to an alternative exposure-gap criterion, which showed no clear advantage over the proposed variance-based score. The authors also clarified that the extra cost is confined to offline training and is modest in the few-shot regime, while the added pipeline complexity appears justified by the consistent gains over simpler augmentations. I am therefore maintaining my score of 5.

**Key Questions For Authors:**

1.  Have you considered other candidate functions for anchor selection, instead of $\mathcal{V}(x)$? For example, $|(\frac{1}{K}\sum^K_{i\neq k} e_i) - e_k|$, which would be a more direct approximation of the generalization error?

**Limitations:**

yes

**Strengths And Weaknesses:**

**Strengths**

The paper is well written, clearly structured, and easy to follow. The authors provide a clear description of the problem setting, the motivation behind targeted data augmentation, and the intuition behind the proposed approach.

**Clear motivation and well-defined problem setting.** The work addresses an important limitation of modern time series forecasting models: their tendency to overfit under data scarcity. The authors provide a compelling argument that not all training samples contribute equally to generalization, and that data augmentation should focus on regions of the data distribution where model ensembles disagree.

**Strong empirical evaluation across settings.** The method is evaluated across multiple widely used time series benchmarks and several forecasting architectures. The experiments cover both few-shot and standard forecasting scenarios, demonstrating consistent improvements over heuristic and learning-based augmentation baselines. Additional comparisons with foundation models such as TimesFM further strengthen the empirical evaluation and help position the method relative to current large-scale approaches.

**Insightful analysis and ablations.** The paper includes several informative analyses that help clarify the contributions of different design choices. For example, ablation studies investigate the impact of anchor sample ratios, model zoo size, and the reinforcement learning optimization procedure.

**Weaknesses**

**Limited justification for the choice of anchor selection function.** The choice of $\mathcal{V}(x)$ as the function for identifying generalization bottlenecks is not very well motivated: no reference to relevant litterature or theoretical justification is provided for the ERM argument. The authors rely on empirical evidence in Figure 2 showing that excluding high $\mathcal{V}(x)$ samples leads to lower test error than training on the full set in most domains. However this experiment does not address whether $\mathcal{V}(x)$ is the best-in-class function to identify challenging samples. It could help to also consider other candidate functions.


**Additional computational cost.** The method necessitates additional models to be trained to produce the augmentations, which raises the overall computational costs. This is acknowledged by the authors, and further justified by the improved performance over other learning-based data augmentation baselines that do not improve performance as much and take longer to train.

**Complex training pipeline.** Although the method is conceptually well motivated, the overall pipeline introduces multiple components (model zoo training, VMAE pretraining, RL finetuning, and forecasting model training). This complexity may make the method more difficult to adopt in practice compared to simpler augmentation strategies.

Minor comments
- l093 rhs : this statement should be motivated via a citation or reference to a later section in the paper.

---

> ### Author Rebuttal · Authors · 2026-03-31
>
> We appreciate the reviewer's valuable comments.
>
> > W1, Q1. The choice of anchor selection function.
>
> **(1) Disagreement as a proxy for discovering overfit-prone data**
>
> The choice of model-zoo variance $V(x)$ is rooted in the principle of **Query-by-Committee (QBC)** and **Ensemble Uncertainty Estimation**. In few-shot regimes, a "generalization bottleneck" is a region where the model's mapping is under-determined by the sparse training data. Statistically, variance across a committee of models (the model zoo) is a robust and standard proxy for this epistemic uncertainty. While we agree that this is a heuristic in the context of time series, it is a well-motivated one: it isolates samples where the Empirical Risk Minimization (ERM) landscape is most sensitive to specific training-data exposure.
>
> **(2) Comparison with the suggested function for anchor selection**
>
> To address whether $V(x)$ is "best-in-class", we follow the reviewer’s suggestion and compare our variance-based criterion with a more targeted exposure-gap function:
> $$V(x)=\left|\left(\frac{1}{K}\sum_{i\neq k}^{K} e_i\right)-e_k\right|.$$
>
> This function explicitly measures the absolute difference between the mean error of models that have seen the sample ($i \neq k$) and the error of the model that has not ($k$). We conduct a head-to-head comparison in MAE/MSE under the same few-shot setting as Table 1:
>
> | Dataset |Original $V(x)$ |  New function ($V_{\text{new}}$) |
> |---|---|---|
> | ETTh1 | **0.423** / 0.402 | 0.425 / **0.399** |
> | ETTh2 | 0.337 / **0.301** | **0.334** / 0.304 |
> | ETTm1 | **0.411** / 0.436 | 0.414 / **0.435** |
> | ETTm2 | 0.273 / **0.194** | **0.271** / 0.197 |
> | Weather | **0.228** / 0.184 | 0.231 / **0.181** |
> | Electricity | 0.255 / **0.165** | **0.252** / 0.169 |
> | Traffic | **0.291** / 0.427 | 0.294 / **0.423** |
> | Exchange | **0.224** / **0.098** | 0.226 / 0.101 |
>
>
> The empirical results show that $V_{\text{new}}$, despite being more explicitly tailored to the seen-vs-unseen contrast, does not yield a clear or consistent advantage over the standard variance $V(x)$. The fact that different disagreement-based functions yield comparable gains suggests that the model-zoo-driven RL algorithm is the true source of ReAugment’s performance, rather than the specific functional form of the scoring metric for anchor selection.
>
>
> > W2. Additional computational cost.
>
> We emphasize that the additional computational cost is strictly confined to the offline augmentation phase. Once the forecasting model is trained with ReAugment, it requires **zero additional computation at inference time**. In practical deployment, where latency and real-time response are critical, ReAugment provides superior accuracy without any speed penalty compared to a model trained on the original data.
>
> Besides, in few-shot scenarios, the primary bottleneck is the **data scarcity and OOD generalization**, not the training time. Since the training sets are small, the absolute overhead is modest (e.g., ~3 minutes for ETTh1). We argue that spending extra minutes during training to "rectify" the training distribution is very worthwhile, especially since simpler, "cheaper" baselines often yield negative performance recovery (Table 3).
>
> Furthermore, as shown in Table 6, while ReAugment is more expensive than heuristic noise, it is significantly **more efficient than generative baselines like TimeGAN** (often 2-3x faster) while consistently delivering stronger accuracy. This demonstrates that our RL-based pipeline is a more targeted and efficient use of compute than traditional adversarial generation.
>
> > W3. Complex training pipeline.
>
> Few-shot forecasting is a fundamentally "under-determined" problem. Simple augmentation strategies (like Gaussian noise) fail because they lack task-awareness. Our pipeline's complexity is what enables the system to: (1) Identify exactly where the model is struggling (Model Zoo), and (2) Produce forecasting-aware data augmentations (VMAE + RL). Each component is essential to bridge the generalization gap that simpler methods cannot reach.
>
> Although the pipeline has several components, it is entirely automated and requires minimal human intervention. To ensure ease of practical use, we maintain a fixed hyperparameter setting ($\eta=0.01$) across all eight datasets, showing that the system does not require per-dataset "manual tuning".
>
> Finally, while we acknowledge that ReAugment involves more stages than heuristic augmentations, a "simpler" but ineffective augmentation that leads to negative performance recovery (Table 3) is a poor substitute for a more structured method that consistently improves performance in sparse-data regimes.

---

> > ### Author Rebuttal · Reviewer_mboD · 2026-04-03
> >
> > The authors have addressed my questions, I have no further concerns.

---

> > > ### Author Response · Authors · 2026-04-04
> > >
> > > We are truly grateful for your supportive and insightful comments, which have been invaluable in enhancing the quality of our manuscript.

---

### Official Review · Reviewer_QsyA · 2026-03-13

**Soundness:** 3
**Presentation:** 2
**Significance:** 3
**Originality:** 3
**Overall Recommendation:** 4
**Confidence:** 4

**Summary:**

ReAugment proposes a targeted augmentation framework for few-shot time series forecasting, based on the intuition that not all training samples should be augmented equally; instead, augmentation should focus on samples around which multiple forecasting models exhibit particularly unstable behavior. To achieve this, the method first uses a model zoo to quantify sample-wise instability, then treats the identified samples as anchors to train a VMAE-based generator, and finally fine-tunes the generation policy with RL (GRPO) so that the augmented samples are more helpful for downstream forecasting performance. Overall, the method forms an “identify–augment–forecast” pipeline, where the final forecasting model is trained on the combination of original and augmented data to better address data scarcity.

**Compliance With Llm Reviewing Policy:**

Affirmed.

**Final Justification:**

I recommend "weak accept" having no additional remarks.

**Key Questions For Authors:**

(1) What additional value does variance-based anchor selection provide beyond simpler alternatives based on error magnitude? Specifically, the paper would be much stronger with a clearer analysis of whether augmentation should target (i) samples that are both high-variance and high-error, (ii) high-variance but low-error samples that reflect instability rather than intrinsic difficulty, or (iii) both.
Without such analysis, it remains unclear whether the proposed criterion captures something genuinely beyond generic hard-sample selection. Explicit comparisons against simpler baselines, such as random or high-error anchors, would further help establish that the reported gains are specific to the proposed variance-based design rather than a generic consequence of prioritizing difficult samples.

(2) Could the authors clarify the fold-construction protocol for the model zoo? If the folds are temporally contiguous, how do you rule out the possibility that high model-zoo variance mainly reflects temporal regime differences or localized distribution shift, rather than true overfit-prone samples?

**Limitations:**

yes

**Strengths And Weaknesses:**

Strengths
- The paper clearly identifies an important limitation of existing augmentation methods: many rely on static heuristics or remain decoupled from the downstream forecasting task. In contrast, it proposes to view augmentation as a task-oriented, closed-loop optimization problem. In this sense, the method is appealing because it does not aim to simply "augment more",  but rather to generate augmentations that are explicitly useful for improving forecasting performance.
- Another strength is that the paper attempts to address not only how to augment, but also what should be augmented. Specifically, the separation between anchor selection and augmentation policy learning gives the overall framework a structurally clean and well-motivated design.

Weaknesses
- In stage A, the proposed method relies heavily on model-zoo variance as the sole criterion for anchor selection. While high variance may reflect sample-wise instability, this statistic discards other potentially important signals such as the absolute error level, making it unclear whether the selected samples are truly hard in terms of forecastability, genuinely overfit-prone, or simply inconsistent across folds. The preliminary findings (Figure 2 and Appendix F) used to justify this choice seem somewhat ambiguous. As a counterexample to the claim around L124, a high variance does not necessarily reflect a clean seen-vs-unseen discrepancy; it may instead arise from disagreement among the seen models themselves. For example, with K=4, a sample with seen-model errors [0.10,0.22,0.35] and an unseen-model error of 0.24 could still have a relatively large variance, even though the unseen model is not particularly worse than the seen ones. This suggests that the current statistic conflates qualitatively different cases, and therefore deserves more fine-grained analysis.
- Some implementation details are left under-specified. The paper states that the training set is partitioned into K disjoint folds to construct the model zoo, but it does not clarify whether these folds are temporally contiguous or randomly shuffled. If the folds are temporally contiguous, the resulting variance may be confounded with regime shifts or temporal location rather than true overfit propensity. If they are shuffled, overlap among nearby sliding-window samples could weaken the intended seen-vs-unseen contrast. The prediction error L(f(x),y) used to define the model-zoo variance is described only as “e.g., MSE,” so it would be helpful to specify the exact loss used for anchor ranking more explicitly.

---

> ### Author Rebuttal · Authors · 2026-03-31
>
> We appreciate the reviewer's valuable comments.
>
> > W1. Justification for model-zoo variance in anchor selection.
>
> While the reviewer’s provided counterexample is theoretically possible, our empirical analysis suggests it does not represent the dominant behavior in few-shot forecasting, and the primary source of variance is the "seen-unseen discrepancy".
>
> In our $K$-fold setup, for any given sample in fold $\mathcal{D}_k$, exactly one model (the held-out model) has not been exposed to that sample or its local temporal patterns, while the remaining $K-1$ models have. In the few-shot regime, this usually leads to a clear **seen–unseen** error gap: seen models tend to fit the training sample better, whereas the unseen model is more likely to incur larger error. Under this condition, the variance in Eq. (1) mainly captures the discrepancy between models that **have seen** the sample and those that **have not**, rather than arbitrary disagreement among seen models. By contrast, absolute error alone cannot distinguish uniformly hard/noisy samples from samples whose prediction is specifically sensitive to training-data exposure.
>
> To substantiate this, we report the average MSE of training samples when evaluated by "Seen" models versus "Unseen" models across all benchmarks:
>
> | Dataset | Seen MSE | Unseen MSE | Gap ($\Delta$) |
> | -- | --| --| -|
> | ETTh1 | 0.259 |0.384 | +48.3% |
> | ETTh2 |    0.228 |0.300 | +31.6%|
> | ETTm1  |    0.326 |0.342 | +4.9%          |
> | ETTm2 |    0.164 | 0.181 | +10.4%         |
> | Weather |    0.153 |  0.178 | +16.3%         |
> | Electricity |  0.120 |      0.149 | +24.2%         |
> | Traffic |    0.356 |      0.391 | +9.8%          |
> | Exchange    |    0.075 |      0.087 | +16.0%         |
>
>
> > W2, Q2. Some implementation details are left under-specified, e.g., the fold-construction protocol and the exact loss used for anchor ranking.
>
> **(1) Fold-construction protocol**
>
> In Stage A, the $K$ folds are **temporally contiguous blocks** of the training period rather than randomly shuffled subsets. As the reviewer correctly noted, random shuffling would cause significant overlap between sliding windows (due to shared observations in the input/output horizons), which would lead to data leakage and artificially deflate the seen-vs-unseen variance. Contiguous partitioning is essential to maintain the integrity of the "exposure sensitivity" signal.
>
> The reviewer raises an insightful point regarding regime shifts of the chronological partitioning. In the few-shot regime, we argue that sensitivity to local temporal characteristics (whether due to distribution shifts or specific patterns) is precisely what constitutes a generalization bottleneck. If a model zoo exhibits high variance on a particular segment, it indicates that the forecasting backbone's performance is highly sensitive to whether it has "seen" that specific temporal regime. Thus, our variance signal intentionally captures these exposure-sensitive regions to prioritize them for augmentation, rather than attempting to isolate a regime-invariant difficulty.
>
> **(2) The loss for anchor ranking**
>
> For anchor ranking, $L(f(x),y)$ is the **sample-wise MSE**, i.e., the mean squared error computed on that sample’s prediction window:
> $$L(f(x), y) = \frac{1}{T} \sum_{t=1}^{T} (\hat{y}_t - y_t)^2$$
> where $T$ is the prediction horizon. The variance in Eq. (1) is then computed over these $K$ scalar MSE values across the model zoo. We will update the manuscript to make this definition explicit.
>
>
> > Q1. What additional value does variance-based anchor selection provide beyond simpler alternatives based on error magnitude?
>
> We appreciate the reviewer’s suggestion to disentangle the effects of variance and absolute error. To address whether our criterion captures something beyond "generic hard-sample selection", we conduct an ablation study within the **top 50% high-variance pool**. We split this pool into two halves based on error magnitude: **High-Var  + 50%-High-error** and **High-Var + 50%-Low-Error**, and compare them against the full High-Variance set, keeping the total volume of augmented data identical.
>
> The results under the few-shot setting (MAE/MSE) are summarized below. The results indicate that, once the anchor pool is restricted to high-variance samples, further filtering by absolute error does not show a clear or consistent advantage. This suggests that the useful signal is primarily carried by **variance-based exposure sensitivity**, rather than by error magnitude alone.
>
> | Dataset | High-var + High-error | High-var + Low-error  | High-var All |
> |---|---|---|---|
> | ETTh1 | 0.427 / 0.406 | 0.425 / 0.404 | 0.423 / 0.402 |
> | ETTh2 | 0.341 / 0.304 | 0.339 / 0.302 | 0.337 / 0.301 |
> | ETTm1 | 0.414 / 0.440 | 0.412 / 0.438 | 0.411 / 0.436 |
> | ETTm2 | 0.277 / 0.198 | 0.274 / 0.196 | 0.273 / 0.194 |
> | Weather | 0.226 / 0.182 | 0.229 / 0.186 | 0.228 / 0.184 |
> | Traffic | 0.291 / 0.430 | 0.293 / 0.429 | 0.291 / 0.427 |

---

> > ### Author Rebuttal · Reviewer_QsyA · 2026-04-04
> >
> > I thank the authors for their responses and additional experiments. I have no additional remarks.

---

> > > ### Author Response · Authors · 2026-04-04
> > >
> > > We sincerely appreciate your careful review and constructive suggestions, which have greatly contributed to the improvement of our paper.

---

### Decision · Program_Chairs · 2026-04-30

**Decision:**

Accept (regular)

**Comment:**

This paper proposes ReAugment, a targeted data augmentation framework for few-shot time series forecasting that combines model-zoo-based sample selection with reinforcement learning-based augmentation policy learning. The key idea is to identify “overfit-prone” regions via cross-model variance and to focus augmentation on these regions in a closed-loop, task-aware manner.

Summary of reviews and discussion.
The reviewers agree that the paper addresses an important and timely problem—data scarcity and overfitting in time series forecasting—and that the proposed framework is technically sound with consistent empirical improvements across multiple datasets and forecasting architectures. Two reviewers (QsyA, mboD) lean clearly positive (weak accept / accept), while one reviewer (KCAc) remains at weak accept and one reviewer (zRAT) is more skeptical, primarily due to concerns about novelty, pipeline complexity, and robustness.

Importantly, after the rebuttal:

1. All reviewers acknowledged that their main technical concerns were largely addressed, with two explicitly stating concerns were fully resolved.
2. Additional experiments and clarifications (e.g., alternative anchor selection criteria, recent baselines, efficiency analysis) strengthened the empirical case.
3. The remaining concerns are primarily about positioning (novelty) and practicality, rather than correctness.

Strengths.

1. Well-motivated problem and formulation. The shift from “how to augment” to “where to augment” is conceptually meaningful and relevant to few-shot regimes.
2. Technically solid and modular framework. The identify–augment–forecast pipeline is clearly structured and supported by ablations.
3. Strong and consistent empirical performance. The method shows improvements across multiple datasets, architectures, and both few-shot and standard settings.
4. Improved clarity after rebuttal. Key concerns regarding anchor selection, efficiency, and baseline comparisons were addressed with additional experiments.

Weaknesses.

1. Limited conceptual novelty. The approach integrates existing components (ensembles, RL, generative models) rather than introducing fundamentally new learning principles. This concern remains valid but is partly mitigated by the effective combination and task-specific formulation.
2. Pipeline complexity and computational overhead. While justified and mostly confined to training, the multi-stage design may limit ease of adoption.
3. Limited analysis of robustness and deeper understanding. Some questions remain regarding behavior on highly stochastic data and deeper characterization of selected samples.

Assessment.
Overall, the paper represents a solid and well-executed contribution to time series forecasting and data augmentation. While the novelty is more at the level of system design and integration rather than fundamental theory, the approach is clearly useful, empirically validated, and likely to inspire follow-up work in task-aware augmentation.

Given the consistency of positive empirical evidence, the resolution of most technical concerns during rebuttal, and the general agreement among reviewers on soundness, I recommend acceptance at medium priority.